# Pressurized Extraction as an Opportunity to Recover Antioxidants from Orange Peels: Heat treatment and Nanoemulsion Design for Modulating Oxidative Stress

**DOI:** 10.3390/molecules26195928

**Published:** 2021-09-30

**Authors:** Lucía Castro-Vázquez, María Victoria Lozano, Virginia Rodríguez-Robledo, Joaquín González-Fuentes, Pilar Marcos, Noemí Villaseca, Maria Mar Arroyo-Jiménez, Manuel J. Santander-Ortega

**Affiliations:** 1Analytical Chemistry and Food Technology Area, Faculty of Pharmacy, University of Castilla-La Mancha (UCLM), Avda. Doctor Jose María Sanchez Ibañez. S/N c.p., 02008 Albacete, Spain; mvictoria.lozano@uclm.es (M.V.L.); virginia.rrobledo@uclm.es (V.R.-R.); joaquin.gfuentes@uclm.es (J.G.-F.); pilar.marcos@uclm.es (P.M.); noemi.vgonzalez@uclm.es (N.V.); mariamar.arroyo@uclm.es (M.M.A.-J.); 2Pharmaceutical Technology Area, Faculty of Pharmacy, University of Castilla-La Mancha (UCLM), Avda. Doctor Jose María Sanchez Ibañez. S/N c.p., 02008 Albacete, Spain

**Keywords:** pressure-extraction, phenolic compounds, flavonoids, volatile compounds, GC-MS, HPLC-MS, antioxidant activity, circular economy, Caco-2, oxidative stress

## Abstract

Orange peel by-products generated in the food industry are an important source of value-added compounds that can be potentially reused. In the current research, the effect of oven-drying (50–70 °C) and freeze-drying on the bioactive compounds and antioxidant potential from Navelina, Salustriana, and Sanguina peel waste was investigated using pressurized extraction (ASE). Sixty volatile components were identified by ASE-GC-MS. The levels of terpene derivatives (sesquitenenes, alcohols, aldehydes, hydrocarbons, and esters) remained practically unaffected among fresh and freeze-dried orange peels, whereas drying at 70 °C caused significative decreases in Navelina, Salustriana, and Sanguina peels. Hesperidin and narirutin were the main flavonoids quantified by HPLC-MS. Freeze-dried Sanguina peels showed the highest levels of total-polyphenols (113.3 mg GAE·g^−1^), total flavonoids (39.0 mg QE·g^−1^), outstanding values of hesperedin (187.6 µg·g^−1^), phenol acids (16.54 mg·g^−1^ DW), and the greatest antioxidant values (DPPH•, FRAP, and ABTS^•+^ assays) in comparison with oven-dried samples and the other varieties. Nanotechnology approaches allowed the formulation of antioxidant-loaded nanoemulsions, stabilized with lecithin, starting from orange peel extracts. Those provided 70–80% of protection against oxidative UV-radiation, also decreasing the ROS levels into the Caco-2 cells. Overall, pressurized extracts from freeze-drying orange peel can be considered a good source of natural antioxidants that could be exploited in food applications for the development of new products of commercial interest.

## 1. Introduction

About one-third of citrus production is industrially processed for orange juice, which generates a huge amount of citrus waste [1]. Citrus peel, the main byproduct of the citrus industry, is a valuable source of bioactive compounds including flavonoids, phenolic acids, tannins, stilbenes, limonoids, coumarins, terpenoids, carotenoids, vitamins, minerals, and dietary fiber [2]. This distinctive profile of bioactive compounds is closely related with diverse biological activities and health-promoting benefits including hypolipidemic, hypoglycemic, anticancerogenic, antibacterial, antifungal, anti-inflammatory, and neuroprotective effects [3,4,5,6,7,8,9] based on their antioxidant ability to reduce oxidative stress [10,11,12,13,14].

Moreover, recent evidence has shown that polyphenols and volatile compounds from citrus peel, with levels around 3–5-fold higher than in citrus juice [15], have a market value for their exploitation [16,17,18] mainly in dried form, although more research is necessary to better understand the effect of different drying techniques on these phytochemicals.

Currently, there is a growing research interest in isolating bioactive compounds from citrus wastes, especially using environmentally friendly technologies [19,20]. Some conventional methods, including liquid-liquid extraction (LLE), solid-liquid extraction (reflux, shaking, stirring, pressing, or heating systems), microwave-assisted extraction (MAE), ultrasound-assisted extraction (UAE), have been used for recovery bioactive compounds from citrus peels [21,22,23,24,25]; nevertheless, sample pre-treatment, processing, and extraction protocols may result in a loss of valuable compounds.

The methods based on the high pressure extraction, such as supercritical fluid extraction (SFE) and pressurized liquid extraction (PLE), have shown many advantages in a green extraction context, reaching the highest isolation efficiency of citrus bioactive compounds [26,27,28]. High pressure extraction constitutes a suitable strategy for the extraction of citrus waste compounds [29] allowing the full exploitation of their properties in potentially new value-added food products [30,31,32,33].

Supercritical fluid extraction (SFE) combines CO_2_ at high temperatures and pressures 20–140 Kpa [34]. In citrus peel residues, SFE has achieved outstanding recovery extraction for hesperidin and narirutin (20 mg·g^−1^ and 2.33 mg·g^−1^, respectively) from mandarin, orange, lemon, and grapefruit waste [35,36].

The use of SFE-CO_2_ can represent a solution for organic solvent elimination and selective extraction of bioactive compounds from vegetal extracts. Indeed, the relative solubilities of the chemical families should be considered, for example using a fractional extraction at increasing pressures, for compounds with low-medium molecular weight [37].

Pressurized water extraction (PLE), also known as accelerated solvent extraction (ASE), is an emerged technique to isolate bioactive compounds using both water and organic solvents in combination with elevated temperature and high pressure (4–20 MPa) [38]. Solvents at subcritical conditions increase the diffusion rate and decrease the viscosity, thus facilitating its matrix penetration, enhancing the contact with the analytes and improving the effective extraction [39]. The pressure extraction with ASE has provided extracts highly enriched in total phenols, terpenoids, polymethoxylated flavones, and phenolic acids and with a high radical scavenging activity in citrus pomaces, grapefruit peel, and mandarin [40,41,42,43,44,45].

At this moment, citrus cast-offs can constitute a new source of antioxidants at low-cost for the food industry in order to develop and valorize innovative reformulations, functional foods, dietary supplements, or nutraceuticals as a part of a strategy for sustainability and contributing to the circular economy. However, citrus polyphenols display a low water-solubility, restricting their integration into aqueous-based food matrices [46]. In addition, many polyphenols suffer degradation when they are exposed to oxygen, light, temperatures, and extreme pH [47]; and even after ingestion they can be degraded in the gastrointestinal tract [48].

Several of these obstacles can be solved using technological approaches focused on the encapsulation of bioactive compounds such as colloidal systems that behave like carriers improving the water-dispersibility, stability, and bioaccessibility and also protecting the encapsulated compounds against degradation [49,50,51,52]. In this sense, the use of platforms that protect bioactive molecules from their premature degradation could be a successful strategy to promote their efficient use. Currently, promising improvements have been made to incorporate nanotechnology into food processing techniques [53,54], to enhance the viability of food antioxidants [55,56,57]. Although little literature is available, naringenin nanoemulsions have proven a complete release, increasing the solubility and bioavailability [58]. Recently, emulsions for citrus polymethoxyflavone encapsulation (mandarin, orange, sweet orange, and bergamot oil) have been designed [59].

However, further studies are required to determine the usefulness of colloidal systems from citrus and its gastrointestinal behavior, as well as comparative studies involving different drying techniques and temperatures in citrus peel since this is how these wastes can be reused.

For all these reasons, the main objectives of the present research were as follows: (I) describe the volatile and phenolic profile of three less-studied orange peels: Navelina, Salustriana, and Sanguina, widely consumed in Spain, using pressurized extraction by ASE; (II) elucidate the effect of oven-drying and freeze-drying pre-treatments on the bioactive compounds and antioxidant potential of the orange peels extracts; (III) incorporate nanotechnology to formulate antioxidant-enriched nanoemulsions of orange peel extract estimating its antioxidant behavior; and (IV) study the protective effectiveness of nanoemulsion against induced oxidative stress on Caco-2 cell line.

## 2. Material and Method

### 2.1. Material: Orange Peel and Cell Line

The three oranges varieties Salustriana (*Citrus sinensis* L. Osbeck cv. “Salustriana”), Navelina (*Citrus sinensis* L. Osbeck cv. “Navelina”), and Sanguina (*Citrus x sinensis* var. “Sanguina”), grown in Valencia (Spain), were purchased at Corte Ingles supermarket. These three Spanish orange varieties have a great interest from the point of view of cultivation as well as food technology and sale.

Human colon carcinoma Caco-2 clone type C2BBe1 cell line was obtained from the American Type Culture Collection (ATCC, Manassas, VA, USA). Dulbecco’s Modified Eagle Minimal Essential Medium (DMEM), fetal bovine serum (FBS), heat-inactivated HyClone, penicillin-streptomicin (PEST), Hanks Balanced Salt Solution (HBSS), phosphate buffer saline (PBS), trypsin-EDTA (0.05%), and non-essential aminoacids (NEAA) were purchased from Labclinics (Barcelona, Spain). Dimethyl sulphoxide (DMSO), triton-X 100, and dichlorofluorescein diacetate (DCFDA) were purchased from Sigma-Aldrich (Madrid, Spain).

### 2.2. Heat Treatment: Oven-Drying and Freeze-Drying of Orange Peels 

Oranges were cleaned with distilled water in the laboratory, and they were immediately peeled. Orange peels from Salustriana, Navelina, and Sanguina were cut into pieces (sized approximately 0.5 × 0.5 cm thick). Sliced peel was divided into four portions: (i) one fresh peel portion to be directly analyzed; (ii) two fractions which were oven-dried at 50 °C and 70 °C, respectively, until their water content was between 9 and 12%; and (iii) one orange peel fraction which was freeze-dried in a vacuum (2.4 × 10^−2^ mB) for 24 h, previously frozen at −78 °C for 12 h, with a condenser temperature of −49 °C.

All biomass resulting from (i), (ii), and (iii) for every variety of orange peel was extracted by ASE and the resulting extracts were analyzed by HPLC-MS and GC-MS to determine volatile compounds, individual flavonoids, phenolic acids, and antioxidant potential.

### 2.3. Isolation of Volatile Compounds by ASE and GC-MS Analysis

Isolation of volatile compounds was performed using of pressurized liquid extraction with an accelerated solvent extractor ASE 200 (Dionex Corp., Sunnyvale, CA, USA). Samples consisted of 10 g of fresh orange peels without albedo (Salustriana, Navelina, and Sanguina) and moisture 68–73%; and 10 g of freeze-dried and 10 g of oven-dried samples of the three varieties (moisture between 5–9%). Extractions were carried out using optimized conditions. Dichloromethane was used as the extraction solvent at 60 °C, and two cycles of 10 min each were carried out under 1500 psi (10.34 MPa) of pressure. The complete system was then rinsed to avoid any carry-over. Next, 50 µL of 4-nonanol solution (1 g/L) was added as an internal standard [60]. Extracts were concentrated under nitrogen gas flow to a final volume of 200 µL and then stored at −20 °C until GC-MS analysis was performed. Extractions were carried out in duplicate.

An Agilent 6890 N gas chromatograph coupled to a 5973 Inert mass selective detector in electron impact mode at 70 eV was used to carry out the identification and quantification of volatile compounds. One microliter (1 μL) of extracts were injected in spitless mode (0.6 min) on a polyethylene glycol capillary column BP-21 (50 m × 0.32 mm × 0.25 μm of film thickness). The oven temperature program was: 60 °C (3 min) − 2 °C/min − 200 °C (30 min). Helium was used as carried gas at a flow rate of 0.8 mL/min^−1^. Injector and transfer line temperatures were 250 °C and 280 °C, respectively. Mass detector conditions were electronic impact (EI) mode at 70 eV; mass acquisition range: 40–450 amu. Identification of the volatile components was performed comparing their GC Kovats index (KI) and mass spectra with the pure standards compounds from Sigma-Aldrich when they were available. Peak identifications were based on comparison with spectral data and retention indices from pure standard compounds when they were available; otherwise, the Wiley G 1035 spectrum library was used. Semiquantitative analyses were carried out assuming a response factor equal to 1 for all the compounds.

### 2.4. Isolation of Individual Flavonoids by ASE and HPLC-DAD-ESI-MS Analysis

The isolation of Salustriana, Navelina, and Sanguina peels was carried out by means of an accelerated solvent extractor ASE 200 (Dionex Corp, Sunnyuale, CA, USA). The ASE-extraction parameters were optimized in previous works [45]: “extractions were performed using 5 g of orange peel which was placed into inox extraction cells of 22 mL. Every cell was filled with methanol and raised to 60 °C. Then, two static extraction phases lasting 10 min was carried out under 1500 psi (10.34 MPa)”. Between extractions, a rinse of the complete system was performed to avoid any carry-over.

Extracts were evaporated using a rotavapor with a vacuum controller (Heidolph, Schwabach, Germany) at 40 °C. Samples were redissolved with 5 mL of methanol, and they were filtered through a Whatman No. 1 filter paper. Samples were kept at −20 °C prior to being used to determine antioxidant activity and phenolic compounds.

The separation, identification, and quantification of phenolic compounds were performed as previously described Castro-Vazquez et al. [45] “by HPLC-DAD-ESI-MS on an Agilent 1100 series system (Agilent, Waldbronn, Germany), equipped with a DAD photodiode detector (G1315B) and a LC/MSD Trap VL (G2445C VL) electrospray ionization mass spectrometry (ESI/MSn) system, both coupled to an Agilent Chem Station (version B.01.03) for data processing.

The samples after filtration (0.20 μm, polyester membrane, Chromafil PET 20/25, Machery-Nagel, Düren, Germany) were injected in duplicate on a reversed-phase narrow-bore column Zorbax Eclipse XDB-C18 (2.1 × 150 mm; 3.5 μm particle; Agilent) protected by a guard column Zorbax Eclipse XDB-C8 (2.1 × 12.5 mm; 5 μm particle; Agilent), both thermostated at 40 °C.

The solvents were as follows: solvent A (acetonitrile/water/formic acid, 3:88.5:8.5, *v*/*v*/*v*), solvent B (acetonitrile/water/formic acid, 50:41.5:8.5, *v*/*v*/*v*), and solvent C (methanol/water/formic acid, 90:1.5:8.5, *v*/*v*/*v*). The flow rate was 0.190 mL/min. The linear solvents gradient was as follows: zero min, 99% A and 1% B; 8 min, 97% A and 3% B; 37 min, 70% A, 17% B, and 13% C; 40 min, 50% A, 30% B, and 20% C; 51 min, 10% A, 40% B, and 50% C; 56 min, 50% B and 50% C; 59 min, 50% B and 50% C; and 65 min, 99% A and 1% B.

For identification, ESI-MSn was used in both positive and negative modes, setting the following parameters: dry gas, N_2_, 11 mL/min; drying temperature, 350 °C; nebulizer, 65 psi; capillary, −2500 V (positive ionization mode) up to 42 min and +2500 V (negative ionization mode) until the end of the chromatogram; target mass, 600 *m*/*z*; compound stability, 40% (negative ionization mode) and 100% (positive ionization mode); trap drive level, 100%; and scan range, 50–1200 *m*/*z*”.

The identification of individual flavonoid compounds was carried out by comparing their retention times and mass spectra provided by the pure standards (from Sigma St. Louis, MO, USA). This was the case for hesperidin, naringin, naringenin, nobiletin, and tangeretin. The identification of compounds for which the standards were not available was performed by comparing the UV spectra, and the [M + H]^+^, [M − H ]^−^ *m*/*z*, with those reported in the literature. These were rutin, neohesperidin, sinensetin, and quercetogen.

Quantification was made by means of external standard calibration lines and were expressed as milligrams of compounds per gram of dry weight (DW). Quantitative results for compounds without chemical standards were expressed in mg naringin equivalents x g^−^^1^.

### 2.5. Determination of Phenolic Acids

The phenolic acid of orange peel extracts was determined according to previous procedure [61]. Standards of phenolic acids were acquired from Sigma (St. Louis, MO, USA) and Fluka (Buchs, Switzerland). HPLC separation, identification, and quantification of phenolic acids were performed on an Agilent 1100 series system (Agilent, Waldbronn, Germany), equipped with a DAD photodiode detector (G1315B) and a LC/MSD Trap VL (G2445C VL) electrospray ionization mass spectrometry (ESI/MSn) system, both coupled to an Agilent Chem Station (version B.01.03) for data processing. After filtration (0.20 µm, polyester membrane, Chromafil PET 20/25, Machery-Nagel, Düren, Germany), 50 µL samples were injected in duplicate, onto a reversed-phase column Zorbax Eclipse XDB-C18 (4.6 × 250 mm; 5 µm particle), thermostated at 40 °C.

The solvents were water/formic acid (999:1 *v*:*v*:*v*) as solvent A; and MeOH/formic acid (999:1 *v*:*v*:*v*) as solvent B. The flow rate was 0.70 mL min^−1^. The linear gradient for solvent B was as follows: 0 min, 5%; 15 min, 35%; 30 min, 43%; 32 min, 100%; 40 min, 5%.

Quantification was made using the DAD chromatograms obtained at 320 nm, respectively, by means of external standard calibration curves. The identity of each compound was established by comparing the retention time, UV-Vis spectra and mass spectra of the peaks in every sample with those previously obtained by injection of standards. For identification, ESI-MSn was used, setting the following parameters: positive ion mode; dry gas, N_2_, 11 mL min^−1^; drying temperature, 350 °C; nebulizer, 65 psi; capillary, −2500 V; capillary exit offset, 70 V; skimmer 1, 20 V; skimmer 2, 6 V; and scan range, 50–1200 *m*/*z*.

### 2.6. Determination of Total Phenolic Content

The total phenol content (TPC) of orange peel extracts was determined according to the Folin-Ciocalteu procedure [62]. Thus, 1.8 mL of deionized water was added to 0.2 mL of each orange peel extract. Then, 0.2 mL of Folin-Ciocalteu reagent was added, and tubes were shaken vigorously. After 3 min, 0.4 mL sodium carbonate solution (35% *w*/*v*) was added together with 1.4 mL of deionized water. Samples were mixed and left in the dark for 1 h. The absorbance was measured at 725 nm using a UV-vis spectrophotometer (Lambda 5, Perkin-Elmer, Seer Green, UK) and the results were expressed in gallic acid equivalents, GAE, using a gallic acid standard curve. Extracts were further diluted if the absorbance value measured above the linear range. Results were expressed as mg GAE x ·g^−1^ respective to dry weight (DW).

### 2.7. Determination of Total Flavonoids Content

Total flavonoid contents were estimated using the method described by other authors [63,64]. The extract (0.5 mL of 1 mg/mL) was mixed with 1.5 mL of methanol. To this mixture, 0.1 mL of 10% aluminum chloride was added, followed by 0.1 mL of 1 M potassium acetate and 2.8 mL of distilled water. The mixture was incubated at room temperature for 30 min. The absorbance was measured by a spectrophotometer at 420 nm. The results were expressed as milligrams quercetin equivalents (QE) per gram of extract (mg QE g^−1^ extract).

### 2.8. Antioxidant Activity of Orange Peels

#### 2.8.1. DPPH Radical Scavenging Assay

The DPPH assay was carried out according to the method proposed by Alañon et al., 2011 [65] and Castro-Vazquez et al. [53]: “where 1,1-diphenyl-2-picrylhydrazyl radical was used as a stable radical. One hundred microliters of different dilutions of extracts were added to 2.9 mL of a 0.06 mM methanol DPPH radical solution. Methanol was used to adjust the zero and the decrease in absorbance was measured at 515 nm every minute for 25 min in a UV-vis spectrophotometer (Helios, Thermo Spectronic, Cambridge, UK). Only values between 20–80% of the initial absorbance of the radical DPPH were taken into consideration. Concentrations were calculated from a calibration curve in the range between 0.1–0.8 mM Trolox. Results were expressed in milligrams of Trolox per gram of dry weight”.

#### 2.8.2. ABTS^•+^ Radical Scavenging Assay

The method used was the ABTS^•+^ decolorization assay in accordance with Alañon et al. 2011 [65] and Castro-Vazquez et al. [53]: “based on the ability of an antioxidant compound to quench the ABTS^•+^ relative to that of a reference antioxidant such as Trolox. A stock solution of ABTS^•+^ radical cation was prepared by mixing ABTS solution and potassium persulfate solution at 7 mM and 2.45 mM final concentration, respectively. The mixture was maintained in the dark at room temperature for 12–16 h before use. The working ABTS^•+^ solution was produced by dilution in ethanol (1:90 *v*/*v*) of the stock solution to achieve an absorbance value of 0.7 at 734 nm. An aliquot of 20 µL of diluted extract was added to ABTS^•+^ working solution. For the blank and standard curve, 20 µL of ethanol or Trolox solution was used, respectively. Absorbance was measured by an UV-vis spectrophotometer at 734 nm immediately after addition and rapid mixing (At = 0) and then every minute for 5 min. Readings at t = 0 min (At = 0) and t = 5 min (At = 5) of reaction were used to calculate the percentage inhibition value for each extract. A standard reference curve was constructed by plotting% inhibition value against Trolox concentration (0.1–0.8 mM). The radical scavenging capacity of extracts was quantified as milligrams of Trolox per gram of dry weight”.

#### 2.8.3. FRAP Assay

The FRAP assay was performed as previously described by Castro-Vazquez et al. [53]: “this spectrophotometric assay measures the ferric reducing ability of antioxidants. The experiment was conducted at 37 °C and pH 3.6. In the FRAP assay, antioxidants present in the extract reduce Fe (III)-tripyridyltriazine complex to the blue ferrous form, with an absorption maximum at 593 nm. The assay was performed by means of an automated microplate reader (Tecan GENios Pro (Tecan Ltd., Dorset, UK)) with 96-well plates. Reagents included 300 mM acetate buffer pH 3.6; 40 mM hydrochloric acid, 10 mM TPTZ solution and 20 mM ferric chloride solution”. The working FRAP reagent was freshly prepared by mixing acetate buffer, TPTZ solution, and ferric chloride solutions in the ratio 10:1:1 and the mixture was incubated at 37 °C. Diluted extract (30 µL) and pre-warmed FRAP reagent (225 µL) were put into each well. The absorbance at time zero and after 4 min was recorded at 593 nm. The calculated difference in absorbance was proportional to the ferric reducing/antioxidant power of the extract. For quantification, a calibration curve of Trolox was prepared. The final results were expressed as milligrams of Trolox per gram of dried orange peel.

### 2.9. Formulation of Nanoemulsion Systems from Sanguina Peel Extracts

Among orange peels, the variety and the heat treatment that prove the highest antioxidant potential was selected in order to be formulated as a lecithin-stabilized nanoemulsion by the solvent displacement technique as previously proposed by Lozano et al. [66]. The selected prototype of orange peel, at the dried condition that provided the best antioxidant potential was chosen as a target for formulation studies and cell assays. Briefly, 63 µL of orange peel extracts was emulsified using 20 mg of lecithin kindly donated by Cargill (Barcelona, Spain), 0.25 mL of ethanol, and 4.75 mL of acetone (Sigma, Spain) under magnetic stirring. The aqueous phase was composed only by 5 mL MilliQ water. After emulsification, the formulations were rota evaporated to a final volume of 5 mL at 37 °C.

### 2.10. In Vitro Studies in Caco-2: Nanoemulsion Viability and Intracellular ROS Levels Measurements

#### 2.10.1. Cell Culture

Caco-2 cells were cultured in DMEM supplemented with 1% (*v*/*v*) PEST, 1% (*v*/*v*) NEAA, 1% (*v*/*v*) L-glutamine, and 10% (*v*/*v*) heat-in-activated FBS in 75 cm^2^ flasks under an atmosphere of 10% CO_2_/90% air at 37 °C. The culture media was replaced every 2 days, and regular passaging was performed by trypsinization. Cells were used between passages x + 24 and x + 30.

#### 2.10.2. Nanoemulsions Effect on Cell Viability

The effect of nanoemulsions on cell viability was studied in Caco-2 cells by the crystal violet assay. Firstly, 2 × 10^4^ cells/well were seeded in 96-well plates and incubated at 37 °C for 24 h. Then, the medium was removed and different aliquots of nanoemulsions and extract dispersed in medium (3–95.5 μL/cm^2^) were added to the cells, which were further maintained at 37 °C for 24 h. After that period of incubation, the formulations were removed, 100 μL of a 0.5% (*w*/*v*) crystal violet staining solution was added to each well and further incubated at 37 °C for 30 min. Cells were washed twice with ultrapure water and left to dry at room temperature for 30 min. Finally, 100 μL of methanol was added to each well and maintained for 30 min with horizontal shaking. Optical density was measured at 570 nm with a plate reader, considering cells treated with DMEM and Triton-X 100 as negative and positive controls.

#### 2.10.3. Intracellular ROS Levels Measurements

The effect of nanoemulsions on the intracellular ROS levels was studied in Caco-2 cells by the dichlorofluorescein (DCFH-DA) assay. Firstly, 2 × 10^4^ cells/well were seeded in 96-well plates and incubated at 37 °C for 24 h. Then, the medium was removed and different aliquots of nanoemulsions and extract dispersed in medium (3–95.5 μL/cm^2^) were added to the cells, which were further maintained at 37 °C for 24 h. Then, cells were washed and 100 μL of DCFH-DA (10 μM) was added and further incubated at 37 °C for 60 min protected from light. After those incubations, cells were washed and 100 μL of H_2_O_2_ 0.3% (*v*/*v*) was added to each well and incubated for 30 min at room temperature protected from light. Supernatants were collected and transferred to black flat-bottom 96-well plates (Cultek, Spain). The measurement of the fluorescent oxidized derivative of DCFH-DA was performed in a plate reader (BMG Labtech’s) using emission and excitation wavelengths of 475 and 125 nm, respectively. The treatments H_2_O_2_ and PBS were considered as positive and negative controls, respectively.

### 2.11. Statistical Analysis

Analysis of variance and multivariate analysis were performed using SPSS 15.0 for Windows statistical package. Differences between chemical data were established for significance at *p* ≤ 0.05 by the Student-Newman-Keuls test. Principal component analysis (PCA) was performed to rate the orange peel samples in groups according to their volatile compounds, phenolic composition, and antioxidant activity.

## 3. Results and Discussion

### 3.1. Volatile Compounds from Orange Peels

Methanol extracts from pressurized ASE isolation were used to quantify the volatile organic compounds (VOCs) by GC-MS. Sixty volatile components were identified in fresh and dried Navelina, Salustriana, and Sanguina peel extracts, among which were sesquitenenes, terpene oxides, terpenoids alcohols, monoterpenoid aldehydes, terpene hydroarbons, terpene esters, chain aldehydes, and C6-alcohols. Table 1 shows results and the relative standard deviation (RSD) expressed in percent and obtained by multiplying the standard deviation by 100 and dividing by the average of two analyzed samples.

In some cases, the formation of new sesquiterpene compounds after drying was observed. Thus, β-farnesene, γ-elemene, and δ-cadinene, absent in fresh Navelina and Salustriana peels (traces), appeared in dried samples (FD, OD-50 °C and OD-70 °C) as can be seen in Table 1. Similar findings were reported by other researchers [67,68,69], probably caused by reactions of oxidation, hydrolysis of glycosylated forms, or rupture of cell walls conditioned by the selected drying technique.

Regarding terpenoid alcohols, both fresh and dried orange peels displayed a wide number of these VOCs and outstanding concentrations of linalool, α-terpineol, nerol, and geraniol, mainly in Navelina peels (Table 1), allowing its differentiation as markers of *Citrus sinensis L*. Osbeck cv. “Navelina”. It is interesting to note that the global levels of terpenoids alcohols quantified in Navelina, Salustriana, and Sanguina peels remained practically unaffected between fresh and freeze-dried samples. However, the effects of drying at 50 °C on their individual terpenoids alcohols levels were variable (Table 1); thus, e.g., *p*-mentha-2.8-dien-1-ol (isomer I) and linalool significantly increased (*p* < 0.05) at OD-D-50 °C, attributed to the hydrolysis of the glycosidically form and releases of their aglycones [69,70]. After OD-70 °C treatment, most of the terpenoids alcohols significantly decreased (*p* < 0.05), caused by thermal degradation and oxidation reaction during drying, e.g., (*Z*)-carveol vary from 2.28 in fresh Navelina peel to 00.03 µg/g after 70 °C.

Monoterpenoid aldehydes were also affected by the drying conditions. Thus, the temperatures not only triggered the formation of some VOCs, they also caused the elimination of others like some monoterpenoids aldehydes (citronellal, perilla aldehyde), which occur in fresh Navelina and Sanguina peel extracts and were absent in samples oven-dried at 70 °C. The content of geranial at OD-50 °C and 70 °C tended to decrease while the content of neral tended to increase upon these treatments. This behavior was previously described [71] as isomerization of geranial, that is *trans*-isomer of citral into *cis*-isomer (neral) as consequence of the drying conditions.

The most interesting finding of the current experiment was the quite similar volatile profile found between fresh and FD orange peels. This fact denotes that the freeze-drying treatment retained as much as possible, better than oven-drying, the initial volatile composition of Navelina, Salustriana, and Sanguina peels. In some cases, freeze-drying even improve the total content that occurred in fresh samples, specifically for sesquitenenes, terpene oxide, and terpene alcohols (Table 1). These results can be explained based on the low processing temperatures applied that avoid the thermal degradation reactions retaining the volatile compounds.

Regarding the amounts of the volatile compounds, terpene hydrocarbons was the chemical family of VOCs with the highest concentration, both in fresh and dried Navelina, Salustriana, and Sanguina peels. Limonene was the major aroma constituent (83.28–92.57%), according with several research focused on orange peels from *Citrus sinensis* [67,72].

Terpene hydrocarbons showed significantly higher levels (*p* < 0.05) in orange peels at OD-50 °C than in the fresh ones since they are categorized as nonpolar compounds. For instance, Salustriana peel showed 2068.1 µg/g and 1715.16 µg/g, respectively (Table 1). The increases in terpene hydrocarbons at OD-50 °C might have been caused by the release of bound volatiles (enzymatic or acid hydrolysis) during the drying process, since volatiles exist in free forms or bound to other molecules such as sugars forming glycosides [73]. Nevertheless, the oven-drying at 70 °C caused a significant decrease (*p* < 0.05) in the global levels terpene hydrocarbons (Figure 1) since drying at medium-high temperatures can cause stripping processes, oxidation, and thermal degradation reactions [74]. In particular, limonene at OD-70 °C decreased to 1000 µg/g in samples dried at 70 °C, even to 852.33 µg/g in Salustriana peels. The reduction of limonene content was also reported in previous research after the drying process at 60–70 °C [75], indicating that samples should not be dried at high temperature.

It is of special note that the rich variety of sesquiterpene in fresh Navelina, Salustriana, and Sanguina peels ranged from 21.76 to 23.58% (Figure 1). An increase was observed of the total sesquiterpene in freeze-dried samples (*p* < 0.05) in comparison with their respective fresh extracts. For instance, Navelina peel showed 76.57 µg/g sesquiterpenes in fresh samples and 86.49 µg/g in freeze-dried peel (Table 1). This increase could be attributed to the breaking of the cells in which the compounds are stored, causing a more effective extraction [70]. In addition, sesquiterpenes have higher molecular weight and thus, they are less volatile and hardly removed from the plant material. However, they are susceptible to oxidation reaction for extended drying time or temperatures (Chua et al., 2019), which can explain the reduction of sesquiterpene amounts after to OD-50 °C and OD-70 °C.

To identify the potential volatile biomarkers that discriminate the three orange peel varieties studied, a factorial principal component analysis was applied. In relation to sesquiterpenes, Navelina fresh peel (*Citrus sinensis* L. Osbeck cv. “Navelina”) was characterized by its levels of valence (35.43 µg/g), α-sinensal (11.8 µg/g), and β-sinensal (10.95 µg/g) described in citrus matrices [76,77]. Salustriana peel (*Citrus sinensis* L. Osbeck cv. “Salustriana”) was identified based on its higher levels of Germecrene D (3.32 µg/g) and Sanguina peel (*Citrus x sinensis* var. “Sanguina”) focused on the amounts of nootkatone (19.07 µg/g), as can be seen in Table 1. All these compounds can be considered differential sesquiterpenes with a highly significant importance as potential specific biomarkers, quantified in concentrations 1.5–4-fold higher in their respective citrus species. These results are consistent with studies on the characterization of sweet orange peel (*Citrus sinensis*) [72,78], although it is the first approach, as far as the authors know, to separately differentiate Navelina, Sanguina, and Salustriana peels based on the sesquiterpene levels.

As concerns the C6-alcohols, 1-hexanol, and (*Z*)-3-hexen-1-ol, insignificant differences (*p* < 0.05) were observed between concentrations of fresh and FD samples from Navelina, Salustriana, and Sanguina peels, indicating that FD preserved these fresh aromas. Nevertheless, their amounts dropped by around 80% in the OD-50 °C sample and decreased (as far as disappearing) upon OD-70 °C, which indicates that C6-alcohols are more sensitive to the dehydration through oven-drying than to FD. For instance, 1-hexanol showed 10.21 µg/g in fresh Navelina peels, 1.04 µg/g in OD-50 °C, and 0.00 µg/g in samples OD-70 °C (Table 1).

Overall, results indicated that there were significant differences between the VOC profile of fresh and dried orange peels, based on the selected drying treatment. However, freeze-drying seems to be an extremely useful technique retaining the volatile compounds of fresh orange peels.

Previous studies have linked the volatile constituents and antioxidant activities of different citrus matrices as *γ*-terpinene, as well as terpinolene and geraniol, with a strong antioxidant activity in terms of DPPH•, ABTS^•+^ radical-scavenging activity; whereas limonene, which is a major component of many citrus peel extracts, has been described with little effect on radical-scavenging [79]. Thus, taking into account that these compounds occur in Navelina, Salustriana, and Sanguina peel extracts (representing 3–10% of its respective volatile group), this criterion could be useful to select or catgorize orange peel samples with antioxidant purposes.

In general, Figure 1 shows the percentage of every chemical family from Navelina, Salustriana, and Sanguina peels, and the effect of fresh/heat treatment.

### 3.2. Total Phenolic Content

The total phenolic contents (TPC) of peel extracts from Salustriana (*Citrus sinensis* L. Osbeck cv. “Salustriana”), Navelina (*Citrus sinensis* L. Osbeck cv. “Navelina”), and Sanguina (*Citrus x sinensis* var. “Sanguina”) were analyzed in fresh, oven-dried at (OD 50–70 °C), and freeze-dried states. The classification of these orange peel extracts based on the highest TPC index was Sanguina > Navelina > Salustriana, as can be seen in Figure 2.

The TPC levels were variable depending on the drying treatment. Thus, TPC of Navelina, Salustriana, and Sanguina peels extracts dried at 50 °C (30.50, 27.16, 64.36 mg GAE·g^−1^ DW, respectively) were significantly higher (*p* < 0.05) than those found in fresh peels (22.75, 19.55, 41.66 mg GAE·g^−1^ DW). These levels were enhanced with oven-drying at 70 °C, reaching 42.00, 31.27, 83.15 mg GAE·g^−1^ DW, respectively. Our results were similar to those reported by Jeong et al. [67], mentioning enhancements of TPC in *Citrus unshiu* peels as the heating temperature increased. Increases of TPC during drying at 70 °C from their precursors by non-enzymatic interconversion between phenolic molecules have also been reported [80].

At the same time, the highest levels of TPC were noted in freeze-dried (FD) extracts from Navelina, Salustriana, and Sanguina peels; they increased two/three-fold compared to their respective fresh peels (Figure 2). Our results were similar to those reported by Farahmandfar et al. [67] who reported remarkable increases in the TPC levels in FD-Navel peels (*Citrus sinensis L*. Osbeck) in comparison with those oven-dried at 45–60 °C.

Navelina, Salustriana, and Sanguina peels obtained by pressurized ASE extraction yielded TPC values 2–10-fold higher than levels reported in other orange peels varieties employing subcritical water extraction, ethanolic liquid isolation, and hot water extraction [35,44,81,82]. These findings prove that the tandem freeze-drying and pressure isolation (ASE) is probably the more suitable technique with regard to the extraction of phenolic compounds from orange peels.

### 3.3. Phenolic Acids

The predominant phenolic acids of Navelina, Salustriana, and Sanguina peels were ferulic, sinapic, *p*-coumaric, caffeic, and chlorogenic acids, while gallic acid and *p*-hydroxy benzoic acids were the lower (Figure 3). On the whole, Sanguina peel exhibited highest amounts of phenolic acids mainly as ferulic acid (0.63 mg g^−1^), sinapic acid (0.94 mg g^−1^), *p*-coumaric acid (0.84 mg g^−1^), caffeic acid (0.66 mg g^−1^), and chlorogenic acid (0.36 mg g^−1^) (Table 2), in accordance with findings reported by Bocco et al. [83]. Chlorogenic acid, based on chemical structure, is highly linked with antioxidant potential and its presence can be of great relevance in the current extracts.

Regarding heat treatment, fresh and OD-50 °C extracts from Navelina, Salustriana, and Sanguina peels displayed similar individual levels of phenolic acids, comparable with the results found by Deng et al. [84] in several citrus peels heating between 50–60 °C.

Sanguina peels showed the highest levels of ferulic acid, sinapic acid, and caffeic acid. The fresh Sanguina peels displayed 0.94, 0.84, and 0.63 mg g^−1^, respectively. Significant increases (*p* < 0.05) were noted after OD-70 °C (2.86, 2.20, 1.57 mg/g) and they were accentuated in FD samples (2.90, 3.41, 2.14 mg g^−1^, respectively) (Table 2); this fact supposes an increase by around 7–10 folds, compared with fresh Sanguina peels.

Our results suggest a reasonable release of phenolic compounds due to the heating temperature of oven-drying treatment, and an outstanding upward trend in FD extracts from Navelina, Salustriana, and Sanguina peels (Figure 3). Parallel findings reported increases in the phenolic acids’ free fraction of several dried citrus peels, at the same time that decreases in the glycoside bound forms [81,85,86].

The comparison of MAE, conventional solvent extraction, UAE, and ASE with data published in the literature showed that microwave assistant, ultrasound, and accelerated solvent has its own effects on the isolation of individual phenolic acids, total polyphenol content, flavonoids, and antioxidant activity of *Citrus sinensis*. Thus, values of TPC from *Citrus sinensis* peels using MAE extraction were 12.09 mg GAE g^−1^ DW [43] and 9.6 mg GAE g^−1^ DW using conventional methanol extraction [8]; however, with ASE extraction, in our optimized conditions, TPC values reached a ranged between 18.45 and 41.66 mg GAE g^−1^ DW for Navelina and Salustriana peels, respectively, revealing the suitability of accelerated solvent extraction by pressure (ASE).

### 3.4. Total Flavonoids Content

The total flavonoid contents (TFC) of fresh orange peels varied between 12.64 (Navelina), 10.86 (Salustriana), and 23.14 mg QE·g^−1^ DW (Sanguina). Increases of TFC were noted in oven-dried peels, especially at 70 °C (*p* < 0.05) reaching 23.33, 19.37, and 31.19 mg QE·g^−1^ DW, respectively (Figure 4). This effect was previously quantified as two-fold higher in citrus peels dried between 70 and 100 °C in comparison with unheated peels [80,87]. Rafiq et al. [88] explained this TFC increase based on changes in the structure of flavonoids and the formation of low molecular weight phenolic compounds during drying.

Our results revealed that freeze-dried samples showed the highest values of (*p* < 0.05) TFC displaying 29.61, 25.05, and 39.00 mg QE·g^−1^ DW in Navelina, Salustriana, and Sanguina peels, respectively (Figure 4). This conduct is probably due to the reduction of water content (concentrating the TFG), together with the low temperature during freeze-drying that preserved the flavonoids, avoiding their thermal degradation. Therefore, data suggest the suitability of FD treatment couple with pressurized ASE isolation to improve the total flavonoids of orange peel extracts that could be used for the reformulation of functional foods.

### 3.5. Individual Flavonoids of Orange Peel Extracts

The proposed ASE-LC-MS/MS method was applied to the isolation, analysis, and quantification of the individual flavonoids in Navelina, Salustriana, and Sanguina orange peels, allowing evaluating the impact of the drying treatments. The factors concerning pressurized ASE optimization were established in a previous experiment and comprised solvent type, extraction temperature, extraction time, and extraction cycles [53].

A total of ten flavonoids were quantified by HPLC-MS. Four glycosilated flavanones (FGs), namely hesperidin, narirutin, rutin, neohesperidin, and naringin; two flavonoid aglycones, specifically hesperetin and naringenin, and four polimethoxylated flavones (PMFs), namely sinensetin, quercetagenin, nobiletin, and tangeretin. The quantification was done based on the UV-data spectra and [M + H]^+^ [M-H ]^−^ *m*/*z* (Table 3).

The highest individual flavonoid levels were observed in Sanguina followed by Navelina and Salustriana peel extracts. Hesperidin and narirutin were the predominant FGs in all the analyzed orange peels, mainly in fresh Sanguina peels with 202.32 mg g^−1^ and 47.83 mg g^−1^, respectively (Table 4). The levels of hesperidin and narirutin here obtained by pressurized extraction (ASE) resulted higher than those recently reported in blood Sanguinello (*Citrus sinensis* L. Osbeck) [90]) and also higher than those quantified in other orange peels using SWE, alkaline hot water, and MAE extraction [35,43,87,89,91].

Differences in PMF levels (*p* < 0.05) observed among sinensetin, quercetogenin, nobiletin, and tangeretin in fresh Navelina, Salustriana, and Sanguina peel were in the range 1.64–8.53, 0.41–7.95, and 0.73–16.51 mg g^−1^ (Table 4), respectively.

In effect, we noted significant differences among the individual flavonoid levels (FGs, aglycones forms, and PMFs) of fresh Navelina, Salustriana, and Sanguina peels that can be primarily attributed to factors such as orange variety, genetic details, cultivar, agroclimatic conditions, or geographical locations (studied in other research [92]). Nevertheless, our results were higher than those reported in the literature when other isolation techniques were used. In this sense, we therefore attribute these greater flavonoids contents to the suitable election of pressurized liquid extraction technique (ASE) improving the yield of FGs, aglycones, and PMFs by 2–5 times compared to data reported with other classical techniques.

Regarding drying treatments, it was observed that dried conditions modified the flavonoids profile of Navelina, Salustriana, and Sanguina peels extracts. The heating temperature impacted the FGs (rutin, narirutin, hesperidin, and naringin) generating decreases after heating, attributed to the cleaving of glycosylated bond fraction [93]. Therefore, for instance, hesperidin from Sanguina peels at OD-50 °C and OD-70 °C decreased at levels that vary from 92.23 to 114.97 mg g^−1^, respectively. The other FGs showed exactly the same behavior, as can be seen in Table 4.

The opposite evolution was noted in the levels of aglycones and PMFs (hesperetin, naringenin, sinensetin, quercetogenin, nobiletin, and tangeretin) of Navelina, Salustriana, and Sanguina peels, which experienced increases after OD in comparison with the fresh samples. Thus, Sanguina peel reached 12.51, 26.40, and 10.89 mg g^−1^ in peels oven dried at 70 °C, since heating treatment might release some low molecular weight phenolic compounds. Similar results and the same flavonoids evolution have been reported in orange peels dried at 70–100 °C using simply liquid extraction and subcritical water extraction under high Pressure [40,80,93,94].

With respect to freeze-drying, moderate decreases of the FGs fraction of Navelina, Salustriana, and Sanguina peels, with respect to the fresh ones, were noted. For example, rutin, narirutin, hesperidin, and naringin in FD-Salustriana peels displayed 5.17, 42.57, 187.59, and 16.32 mg g^−1^, respectively, which were very similar to those quantified in fresh samples (Table 4). It seems obvious that freeze-drying retains, to a large extent, the initial composition. Data are in a good agreement with those showed by Molina-Calle et al. [95] in lyophilized peels of eight different orange varieties.

Freeze-drying also affected the aglycones and PMFs levels from Navelina, Salustriana, and Sanguina, which significantly increased their levels (*p* < 0.05). The concentration of hesperetin, naringenin, sinensetin, quercetogenin, nobiletin, and tangeretin in freeze-dried samples was around twice that of the fresh peels. For instance, naringenin and nobiletin showed 3.77 and 3.3 mg/g, respectively, in fresh Navelina peel and reached 8.05 and 5.53 mg g^−1^ in FD-Navelina peel (Table 4). The comparison of results with the few data available in the literature were matched [87,96].

Considered all together, these results suggest that Navelina, Salustriana, and Sanguina peels are a good source of phenolic acids and free and bound flavonoids. Additionally, Sanguina peels were the main phenolic-rich extract, followed by Navelina and Salustriana peels.

Freeze-drying seems to be the most suitable drying technique to retain the initial composition. The evaluation of the two methods of water removal revealed that freeze-dying preserved the concentration of the flavonoids, while oven-drying (at 50 °C and 70 °C) produced significant decreases in the FGs (by oxidation reactions due to temperature exposure) and increases of aglycone forms and PMFs.

### 3.6. Effect of Drying Treatments on the Antioxidant Activity of Orange Peels

The antioxidant properties of Navelina, Salustriana, and Sanguina peel extracts were evaluated based on their ability to reduce free radicals formed during oxidation processes, using DPPH•, FRAP, and ABTS^•+^ assays. The effects of oven-drying temperatures (50 °C, 70 °C) and freeze-drying process were also studied, as shown in Table 5.

The first conclusion that can be drawn is the positive effect of freeze-drying in the antioxidant activity of the three orange peel extracts when they were compared with to fresh ones. Thus, for instance, fresh Navelina, Salustriana, and Sanguina peels extracts showed an initial DPPH value of 44.82, 33.62, and 67.24 mg trolox·g^−1^ DW, respectively, whereas freeze-dried samples displayed 71.57, 53.68, and 107.36 mg trolox·g^−1^ DW. This effect linked the freeze-drying process with an increase of redox-active metabolites and phenolic compounds, whcih play an important role in adsorbing and neutralizing free radicals. The same trend was observed for the ABTS and FRAP values in free and FD orange peels.

Similar results were corroborated by previous studies providing the outstanding antioxidant activity of ASE extracts from freeze-dried citrus peels compared with other drying treatments [67,97,98]. Therefore, it could be inferred that freeze-drying is the most efficient drying approach to preserving, even improving, the bioactive compounds and the antioxidant properties of orange peel extracts.

The DPPH values of oven-dried (50 °C, 70 °C) Navelina, Salustriana, and Sanguina peels increased after heat treatment, as can be seen in Table 5, mainly at 70 °C. Significant differences (*p* < 0.05) between fresh and oven-dried extracts were observed. For example, for Sanguina peel extracts, the temperature of 50 °C showed an increase of DPPH from 38.73 mg trolox·g^−1^ DW (fresh) to 48.64 mg trolox·g^−1^ DW (50 °C) and 63.36 mg trolox·g^−1^ DW (70 °C). The trend of these results was consistent with those reported for citrus extracts by other researchers [81,86] and should be attributed to the relationship between increases of dried temperatures and a higher generation of breakdown antioxidant products.

The ABTS^•+^ assay also revealed significant and highest values (*p* < 0.05) of free radical scavenging activity in FD orange peels extracts. It is worth noting that Sanguina freeze-dried peels reached 569.23 mg trolox·g^−1^ DW, whereas its fresh extracts just 202.69 mg trolox·g^−1^ DW.

The drying temperature displayed a profound effect on the antioxidant activity. The ABTS^•+^ activity values of Navelina, Salustriana, and Sanguina peels dried at 50 °C and 70 °C showed an increase when they were compared with fresh extracts, although the rise was less pronounced than in the case of the freeze-dried treatment. For instance, in oven-dried Sanguina peels, the ABTS^•+^ values ranged between 283.48 (50 °C) and 364.58 (70 °C) and 569.23 mg trolox·g^−1^ DW (Table 5). Our results were consistent with those reported by Xu et al. [99] and Jeong et al. [100] who reported significant increases (*p* < 0.05) in ABTS values and newly formed low molecular weight phenols in citrus peel extract after heat treatment in comparison with unheated peels. Therefore, data suggest that a proper heat treatment could be used to enhance the antioxidant capacity of citrus peel by-products. The higher levels of hesperidin and the rest of aglycone flavonoids and PMFs in oven dried samples at 70 °C, vs. 50 °C, also conditioned these results in the three orange peels studied.

Regarding FRAP chelating activity, once again, noteworthy increases in values (*p* < 0.05) for freeze-dried of Navelina, Salustriana, and Sanguina peels were observed. Data obtained vary from 26.89, 18.83, and 38.73 for fresh orange peels to 49.49, 34.64, and 71.26 mg trolox·g^−1^ DW for freeze-dried samples, respectively. This fact suggested that the freeze-drying treatment might produce changes not only for cellular modifications and dissociation or liberation of some phenolic compounds from biological structures, but also the alteration in their chemical structures which could make possible the conversion of insoluble phenols into more soluble forms.

The increase of FRAP values after oven-drying at 50 °C and mainly at 70 °C were observed in Navelina, Salustriana, and Sanguina peels as the temperature increased, which is undoubtedly related to their phenolic content [67]. The chelating activity values of samples dried at 70 °C increased around 1.6 times, reaching 42.92, 30.4, and 63.36 mg trolox·g^−1^ DW in Navelina, Salustriana, and Sanguina peels, respectively.

Taken together with the antioxidant activity values, it seems evident that freeze-dried Salustriana peels could be exploited in the food industry due to their richness in bioactive components that wield antioxidant properties. In the near future, it could be incorporated into a large number of applications as an antioxidant opportunity in circular economy for food industry.

The correlation coefficients between total polyphenol content (TPC), total flavonoids content (TFC), and ABTS, FRAP, DPPH assays are shown in Figure 5. The Pearson correlation coefficients calculated between TPC and antioxidant activity were 0.980, 0.947, and 0.911 for DPPH, FRAP and ABTS, respectively; and 0.956, 0.945, and 0.903 for TFC. Data suggest a strong correlation between the antiradical activity vs. TPC and TFC in the studied orange peels.

A striking and linear relationship between total TPC, TFC, and antioxidant capacity of peel extracts was noted in Navelina, Salustriana, and Sanguina peel extracts, both for fresh and dried samples, showing *R*^2^ correlation coefficient between 0.9707 and 0.9998 (Figure 5).

Extracts with the highest phenolic contents had the highest antioxidant potential in all assayed systems; whilst extracts characterized by smaller total phenolic levels exhibited lower antioxidant capacities. For instance, freeze-dried Salustriana peel extracts which had the highest TPC (117,15 ± 6.64 mg GAE·g−1 DW) and the highest TFC (46.00 ± 0.44 mg QE·g^−1^ DW) also showed the highest ABTS (569.23 ± 6.16 mg trolox·g^−1^ DW), FRAP (71.26 ± 0.43 mg trolox·g^−1^ DW), and DPPH (107.36 ± 0.84 mg trolox·g−1 DW) (Table 5), which means that TPC and TFC were outstanding factors accounting for the antioxidant activity of the orange peel extracts.

### 3.7. Nanoemulsion Systems from Orange Peels

The freeze-dried (FD) Sanguina peel extract showed the highest values of antioxidant activity, TPC, TFC, and flavonoids, which were even higher than those quantified in Navelina and Salustriana peels. Therefore, freeze-dried Sanguina peel extract seems to be the most efficient drying approach. For this reason, this extract was selected to design a nanoemulsion as antioxidant-based nanocarrier that offers oxidative protection to FGs, aglycones forms, and PMFs.

The appropriate methodology selected allowed the successful formulation of colloidal systems as NE-Sanguina peel extract, loaded with antioxidants. The NE systems from Sanguina peels extracts presented a hydrodynamic size value of 130 ± 10 nm and showed a negative surface charge ζ-potential of −41 ± 5 mV, which conferred suitable electrostatic stability to the formulation and polydispersity index (PDI) of 0.410.

Firstly, we evaluated the effect of the formulation process on the antioxidant capacity. Interestingly, the freshly prepared NE-Salustriana peel extracts did not show significant differences (*p* < 0.05) with respect to the non-emulsified extract (Figure 6) in terms of DPPH, FRAP, and ABTS values. These results reveal that the mild conditions of the formulation method did not compromise the antioxidant capacity of the initial extract and correlated with a previous work in which comparable antioxidant activities were obtained both for free and emulsified *Citrus reticulata* peels oil [99,101].

The next experiment was focused on the effect of the emulsification process on the protection against external oxidative stress. Non-emulsified and NE-Sanguina peel extracts were incubated with UV radiation at room temperature for 60 min. The antioxidant parameters were monitored to elucidate their antioxidant potential using the non-oxidized samples, as control. After the oxidation process, less marked declines of antioxidant parameters in NE-Sanguina peel extract (FD) were observed compared with the free Sanguina peel extract (FD). Thus, the DPPH^•^, ABTS^•+^, and FRAP values of NE-Sanguina peel after UV damage were 79.94, 375.74, and 51.20 mg trolox·g^−1^ DW, respectively. However, the non-emulsified Sanguina peel extracts after UV just displayed 47.23, 281.74, and 30.32 mg trolox·g^−1^ DW (Figure 6).

In brief, Sanguina peel extract (FD) retained just 42–50% of the initial antioxidant activity, whereas NE-Sanguina peel extract (FD) retained between 70 and 80% after UV oxidation (Figure 6), thereby enhancing its overall bioactivity and revealing the remarkable antioxidant protection of the NE system in comparison with the free Sanguina peel extract. Based on the previous results of our research group, the higher resistance of the NE-Sanguina peel extracts (FD) against light-induced degradation could be due to the spherical form of the nanoemulsion droplet that minimizes the ratio of extract directly exposed to the external light and then its photochemical oxidation. On the other hand, the scattering and the reflection of the light by nanoemulsion droplets may also contribute to the screening of the light [55,99].

Our results support the effectiveness of NE-Sanguina peel extracts (FD) in retaining the antioxidant activity, protecting the bioactive compounds from degradation when they are exposed to oxidative UV radiation. For this reason, NE-Sanguina peel extracts (FD) could be considered a suitable antioxidant strategy with interesting approaches to consider its incorporation into food, beverage, nutraceutical formulations, or even translate to pharmaceutical industry.

### 3.8. In Vitro Studies in Caco-2: Nanoemulsions’ Effects on Cell Viability

The results obtained in the crystal violet assay indicated that neither free Sanguina peel extracts (FD) nor NE-sanguina peel extracts (FD) produce toxicity in the cell line or compromise its viability. Data showed that formulations of NE-Sanguina peel extract (FD) did not alter the viability of the Caco-2 cells, showing that these nanoemulsions system were safe, in terms of toxicity. It is important to remark that this behavior was kept along the entire range of concentrations/conditions tested, even at the highest, which was barely diluted (Figure 7).

### 3.9. In Vitro Studies in Caco-2: Inhibition of Intracellular ROS Levels

The dichlorofluorescein assay provides information about the intracellular ROS using the fluorescent levels achieved by the oxidation of the DCFH-DA reagent. This study indicates the ability of freeze-dried (FD) Sanguina peel like free Sanguina peel extracts (FD) and the formulations of NE-Sanguina peel extract (FD) in order to protect the Caco-2 cells against an oxidant insult and oxidative stress, as is the incubation with H_2_O_2_ 0.3% (*v*/*v*).

The results showed that the free Sanguina peel extract (FD) decreased ROS levels in a concentration-dependent fashion, showing a 50% reduction at the highest concentration tested. Nevertheless, it was opposite to the results observed for the NE-Sanguina peel extract (FD), with intracellular ROS levels higher than those achieved by H_2_O_2_ itself (Figure 8). This pro-oxidant behavior could be due to a higher intracellular concentration of the formulated NE-Sanguina peel extract (FD) dispersed in the medium when compared with the free Sanguina peel extract (FD). NE as vehicle probably caused a higher internalization of the extract through cellular membrane and therefore a greater transport of antioxidants into the cell, whose concentrations could cause the pro-oxidant effect. This would be in line with previously published results of our group [55] that showed how lecithin-stabilized nanoemulsions were efficiently taken up by Caco-2 cells.

## 4. Conclusions

We can affirm that pressure extraction using ASE provided orange peel extracts with outstanding levels of bioactive compounds.

The effect of drying treatment (freeze-drying and oven-drying) significantly affected the VOC fractions, total phenol composition, flavonoids, and antioxidant activity of Navelina, Salustriana, and Sanguina peel extracts, relative to fresh samples.

Freeze-dried treatment was an appropriate drying method to retain, better than oven-drying, the initial volatile composition of Navelina, Salustriana, and Sanguina peels, even improving the global levels of sesquitenenes, terpene oxides, and terpene alcohols in Navelina, Salustriana, and Sanguina peels. The oven-drying at 70 °C favors the formation of aglycones and PMFs, while the freeze-drying preserved the FG levels, in the three varieties of the three studied *Citrus sinensis* peels.

The highest antioxidant values (DPPH, ABTS, FRAP) were observed in freeze-dried Navelina, Salustriana, and mainly in Sanguina peels, concluding that FD can be considered a tool for increasing the global antioxidant potential of orange peel extracts. The screening of samples identified freeze-dried Sanguina peels as the best choice in terms of enriched-bioactive constituents and antioxidant potential, so it was selected for the nanotechnology approaches. Interestingly, the design of flavonoids-loaded nanoemulsions from Sanguina peel extract provided protection (70–80%) against oxidative UV radiation and decreased the ROS levels in the Caco-2 cell without compromising its viability.

In future, results may open new prospects for the valorization and full exploitation of freeze-dried orange peel that could be potentially be incorporated into food products as new value added with health-promoting properties.

## Figures and Tables

**Figure 1 molecules-26-05928-f001:**
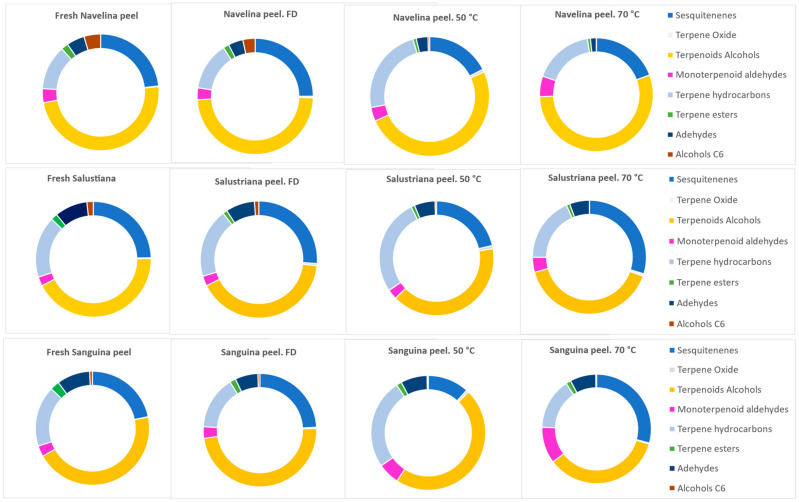
Percentage (%) of every chemical family of VOCs respective to the total of volatile compounds, for every orange variety (Navelina, Salustriana, and Sanguina) and for every treatment: fresh, freeze-dried (FD), and oven-dried (50 °C, 70 °C).

**Figure 2 molecules-26-05928-f002:**
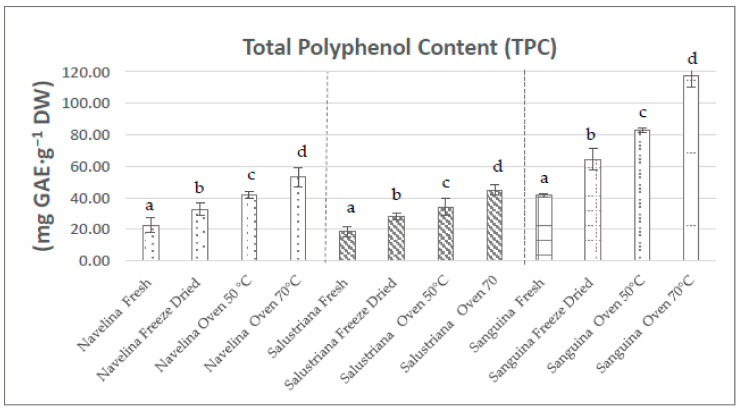
Total phenolic contents of ASE extracts from Navelina, Salustriana, and Sanguina peels, fresh and after different drying treatments. a, b, c, d: Different letters in the same column denote a significant difference according to the Student-Newman-Keuls test, at *p* < 0.05. GAE—gallic acid equivalent.

**Figure 3 molecules-26-05928-f003:**
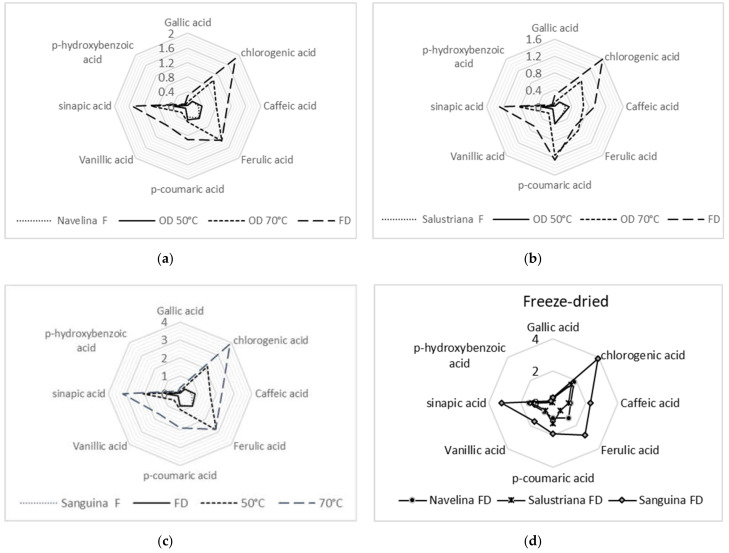
Phenolic acids (mg g^−1^ DW) of Navelina (**a**), Salustriana (**b**), and Sanguina peels (**c**), fresh (F), after freeze-drying (FD), and after oven-drying (OD-50 °C, OD-70 °C). (**d**) shows the comparison between the three FD orange peels.

**Figure 4 molecules-26-05928-f004:**
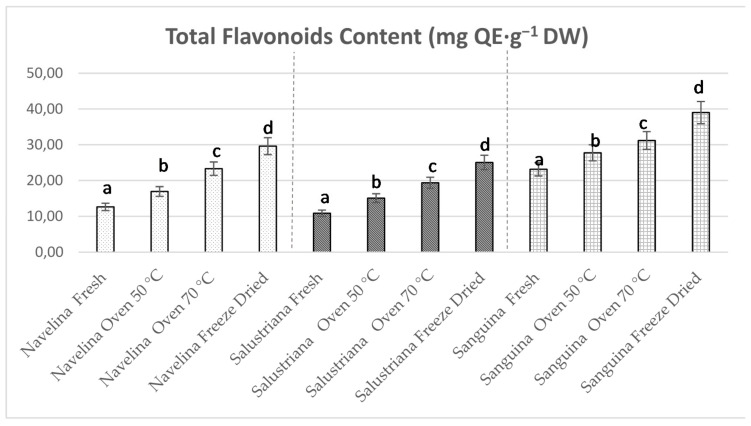
Total flavonoids contents (TFC) from pressurized extracts of Navelina, Salustriana, and Sanguina, both fresh and after drying treatments: oven-dried (50 °C), oven-dried (70 °C), and freeze-dried. QE–quercetin equivalent.

**Figure 5 molecules-26-05928-f005:**
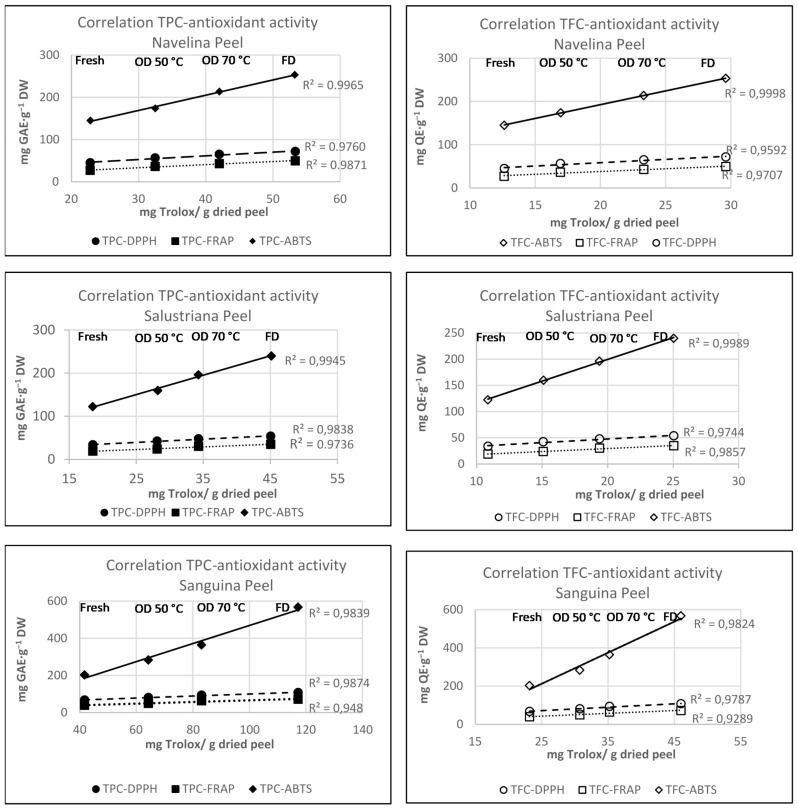
Correlation between total phenols content, total flavonoids content, and DPPH, FRAP and ABTS assay, of fresh, oven-dried, and freeze-dried orange peels from Navelina, Salustriana, and Sanguina varieties. OD—oven-dried; FD—freeze-dried; DW—dry weight; TPC—total polyphenol content (expressed as mg gallic acid equivalent AE·g^−1^ DW); TFC—total flavonoid content (expressed as mg quercetin equivalent·g^−1^ DW).

**Figure 6 molecules-26-05928-f006:**
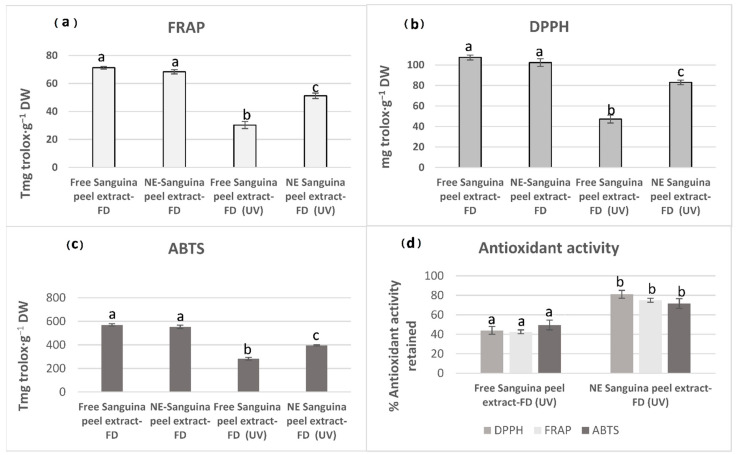
Antioxidant activity DPPH (**a**); FRAP (**b**); and ABTS (**c**) of pressurized extracts from free FD Sanguina peel in comparison with nanoemulsion of Sanguina peels FD, after 60 min of UV lamp oxidation. (**d**) shows the percentage of the antioxidant activity retained by NE vs. free extract after the UV radiation.

**Figure 7 molecules-26-05928-f007:**
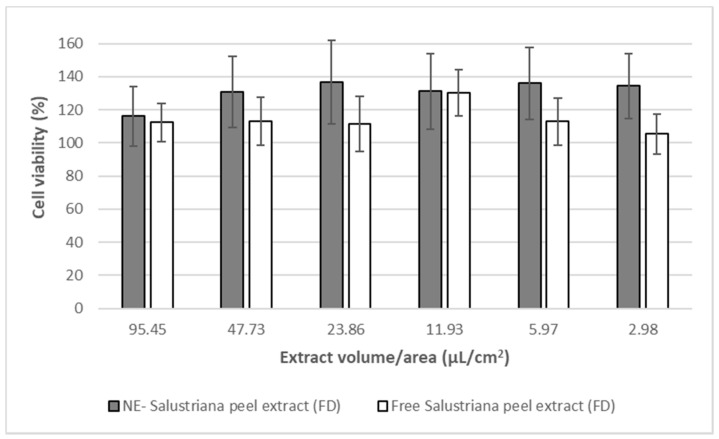
Effect of nanoemulsion (NE) on the cell viability of Caco-2, in comparison with the pressurized free extract. Free Salustriana peel extract freeze-dried, previously freeze-dried (FD). NE-Salustriana peel extract freeze-dried, previously freezedried (FD).

**Figure 8 molecules-26-05928-f008:**
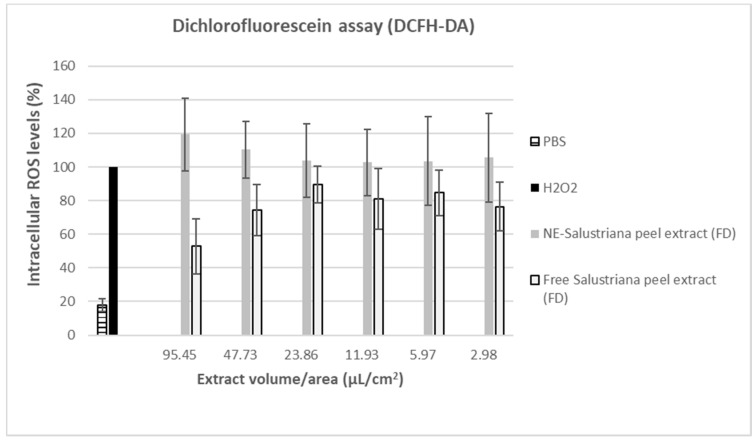
Effect of nanoemulsions of Sanguina peel (lecithin-stabilized) to reduce intracellular ROS generation in Caco-2 cells, after oxidative stress induced by H_2_O_2_. PBS—phosphate buffered saline. Free Salustriana peel extract freeze-dried, previously freeze-dried (FD). NE— Salustriana peel extract freeze-dried, previously freeze dried (FD).

**Table 1 molecules-26-05928-t001:** Mean Concentration (μg. g^−1^ DW) and relative standard deviation (%) of volatile compounds in citrus peel extracts from Navelina (*Citrus sinensis* L. Osbeck cv. “Navelina”), Salustriana (*Citrus sinensis* L. Osbeck cv. “Salustriana”), and Sanguina (*Citrus x sinensis* var. “Sanguina”), fresh, oven-dried (50 °C, 70 °C), and freeze-dried (FD).

KI	COMPOUNDS	Navelina Peel								Salustriana Peel								Sanguina Peel							
		Mean Fresh	RSD	Mean FD	RSD	Mean Oven 50 °C	RSD	Mean Oven 70 °C	RSD	Mean Fresh	RSD	Mean FD	RSD	Mean Oven 50 °C	RSD	Mean Oven 70 °C	RSD	Mean Fresh	RSD	Mean FD	RSD	Mean Oven 50 °C	RSD	Mean Oven 70 °C	RSD
**Sesquitenenes**																									
1476	α-copaene	1.54 ^a^	(7.76)	1.71 ^a^	(10.91)	0.52 ^b^	(8.32)	0.38 ^b^	(12.97)	1.58 ^a^	(14.91)	1.2 ^a^	(16.90)	0.44 ^b^	(0.59)	0.34 ^c^	(7.16)	2.37 ^a^	(2.13)	2.92 ^b^	(5.07) ^a^	1.26 ^c^	(6.93)	1.82 ^d^	(5.75)
1630	γ-elemene	traces		0.29 ^a^	(11.78)	0.26 ^a^	(7.18)	0.71 ^b^	(7.52)	traces		0.56 ^a^	(8.13)	0.36 ^b^	(3.61)	0.57 ^a^	(5.61)	0.33 ^a^	(7.56)	0.73 ^b^	(8.15)	0.13 ^c^	(8.63)	0.27 ^d^	(10.80)
1654	β-farnesene	traces		0.58 ^a^	(9.82)	0.24 ^b^	(3.55)	2.17 ^c^	(15.98)	traces		0.50 ^a^	(0.77)	0.25 ^b^	(0.71)	0.55 ^a^	(16.94)	0.27 ^a^	(10.17)	0.21 ^a^	(5.82)	0.24 ^a^	(9.51)	0.46 ^b^	(13.48)
1701	valencene	35.43 ^a^	(0.98)	36.67 ^a^	(12.97)	30.2 ^b^	(7.82)	13.31 ^c^	(10.29)	21.75 ^a^	(5.06)	27.45 ^b^	(7.76)	25.16 ^ab^	(7.99)	22.63 ^a^	(7.99)	20.54 ^a^	(7.63)	26.4 ^b^	(9.38)	5.86 ^c^	(10.07)	12.58 ^d^	(8.63)
1704	germacrene D	0.44 ^a^	(0.00)	0.81 ^b^	(6.73)	0.44 ^a^	(5.08)	0.12 ^c^	(8.29)	3.32 ^a^	(20.20)	3.37 ^a^	(6.88)	2.31 ^b^	(12.91)	2.04 ^b^	(7.31)	0.11 ^a^	(6.15)	0.38 ^b^	(0.31)	0.05 ^c^	(1.36)	0.21 ^d^	(4.85)
1723	α-farnesene	0.51 ^a^	(0.34)	0.87 ^b^	(12.97)	0.56 ^a^	(0.28)	0.42 ^c^	(4.98)	1.09 ^a^	(0.29)	0.88 ^b^	(11.05)	0.43 ^c^	(7.88)	0.31 ^d^	(0.26)	0.68 ^a^	(0.98)	3.22 ^b^	(8.07)	0.62 ^ac^	(9.51)	0.73 ^c^	(10.16)
1740	δ-cadinene	traces		0.5 ^a^	(11.47)	0.19 ^b^	(6.86)	0.62 ^a^	(11.78)	traces		0.36 ^a^	(3.96)	0.21 ^b^	(4.56)	0.26 ^b^	(3.01)	0.34 ^a^	(5.66)	0.91 ^b^	(2.11)	0.13 ^c^	(10.97)	0.65 ^d^	(6.86)
2016	nerolidol	2.06 ^a^	(8.15)	2.28 ^a^	(11.55)	0.42 ^b^	(5.82)	0.30 ^c^	(6.86)	0.99 ^a^	(7.99)	0.84 ^a^	(3.91)	0.64 ^c^	(4.07)	0.41 ^d^	(5.35)	0.66 ^a^	(7.83)	0.91 ^b^	(6.82)	0.41 ^c^	(12.97)	0.46 ^c^	(2.77)
2236	β-sinensal	11.8 ^a^	(6.86)	14.08 ^b^	(7.44)	5.69 ^c^	(7.52)	3.65 ^d^	(7.52)	3.19 ^a^	(8.63)	3.36 ^a^	(2.00)	2.6 ^b^	(0.01)	2.15 ^c^	(0.28)	5.17 ^a^	(7.76)	7.47 ^b^	(5.70)	0.35 ^c^	(3.85)	1.38 ^d^	(0.98)
2331	α-sinensal	10.95 ^a^	(12.55)	13.77	(14.14) ^a^	4.91 ^b^	(10.05)	3.01 ^c^	(10.05)	4.32 ^a^	(18.69)	4.18 ^a^	(18.21)	1.86 ^b^	(0.01)	1.62 ^b^	(11.39)	4.32 ^a^	(8.32)	5.44 ^b^	(3.33)	0.73 ^c^	(6.67)	1.11 ^d^	(8.07)
2507	nootkatone	5.81 ^a^	(7.82)	6.16 ^a^	(2.59)	4.43 ^b^	(7.56)	2.86 ^b^	(7.56)	9.73 ^a^	(1.22)	11.03 ^b^	(8.13)	6.86 ^c^	(0.78)	4.03 ^d^	(10.07)	19.07 ^a^	(5.70)	27.75 ^b^	(3.53)	10.9 ^c^	(9.59)	13.21 ^d^	(7.25)
2083	elemol	3.93 ^a^	(9.47)	3.47 ^a^	(9.38)	0.81 ^a^	(3.51)	0.62 ^c^	(4.98)	1.82 ^a^	(8.15)	1.81 ^a^	(13.48)	1.81 ^a^	(7.63)	1.29 ^b^	(7.63)	1.3 ^a^	(10.14)	0.88 ^b^	(8.13)	1.65 ^c^	(7.76)	0.51 ^d^	(0.77)
2121	γ-eudesmol	1.12 ^a^	(12.91)	1.17 ^a^	(3.55)	0.25 ^b^	(14.88)	0.78 ^c^	(7.52)	0.67 ^a^	(9.38)	0.54 ^b^	(6.67)	0.51 ^b^	(13.26)	0.33 ^c^	(13.26)	0.21 ^a^	(6.73)	0.31 ^b^	(0.01)	0.32 ^b^	(8.32)	0.32 ^b^	(4.07)
2183	δ-eudesmol	2.15 ^a^	(1.40)	2.21 ^a^	(9.38)	1.17 ^b^	(9.38)	1.27 ^b^	(2.28)	5.05 ^a^	(10.05)	9.09 ^b^	(7.88)	5.09 ^a^	(6.67)	3.83 ^c^	(6.67)	7.81 ^a^	(1.75)	9.51 ^b^	(2.77)	4.68 ^c^	(13.26)	7.57 ^a^	(7.16)
2348	(*E*)-farnesol	1.33 ^a^	(7.63)	1.93 ^b^	(3.63)	1.4 ^a^	(8.63)	1.03 ^c^	(8.07)	1.58 ^a^	(12.55)	1.24 ^ab^	(4.85)	1.08 ^b^	(0.01)	0.68 ^c^	(0.01)	1.04 ^a^	(7.52)	1.62 ^b^	(7.31)	0.97 ^a^	(12.49)	1.02 ^a^	(6.86)
	Total	76.57		86.5		51.49		31.25		55.09		66.41		49.61		41.04		64.22		88.66		28.3		42.3	
**Terpene oxide**																									
1428	(*Z*)-linalool oxide	0.11 ^a^	(7.83)	0.21 ^b^	(6.07)	0.19 ^b^	(4.47)	traces		0.17 ^a^	(8.44)	0.44 ^b^	(8.63)	0.78 ^c^	(5.14)	0.51 ^b^	(6.36)	0.07 ^a^	(10.88)	0.60 ^b^	(7.75)	0.57 ^b^	(3.40)	traces	
1430	(Z)-limonene-oxide	0.31 ^a^	(4.98)	0.64 ^b^	(3.82)	0.74 ^c^	(0.77)	traces		0.11 ^a^	(6.73)	0.79 ^b^	(2.70)	0.91 ^c^	(7.63)	0.42 ^c^	(7.04)	0.09 ^a^	(8.32)	0.32 ^b^	(6.86)	0.41 ^b^	(5.82)	0.20 ^c^	(7.91)
1453	(*E*)-linalool oxide	0.16 ^a^	(1.40)	0.65 ^b^	(6.87)	1.24 ^c^	(2.77)	traces		0.25 ^a^	(22.63)	0.29 ^ab^	(15.48)	0.31 ^b^	(6.96)	0.01 ^c^	(0.00)	0.10 ^a^	(14.14)	0.27 ^b^	(8.63)	0.67 ^c^	(5.08)	0.18 ^ab^	(6.82)
	Total	0.58		1.50		2.17		0.03		0.53		1.52		2.00		0.94		0.26		1.19		1.65		0.39	
**Terpenoids alcohols**																									
1541	linalool	66.79 ^a^	(0.88)	79.79 ^b^	(7.86)	81.02 ^b^	(6.33)	48.74 ^c^	(2.24)	32.52 ^a^	(7.44)	35.17 ^a^	(14.76)	45.12 ^b^	(1.53)	18.47 ^c^	(6.90)	63.90 ^a^	(6.67)	108.33 ^b^	(3.24)	80.96	(1.36)	24.52	(6.99)
1585	α-terpineol	32.08 ^a^	(11.55)	25.01 ^b^	(7.33)	7.81 ^b^	(7.25)	7.78 ^b^	(7.44)	17.61 ^a^	(8.71)	20.42 ^b^	(7.91)	7.61 ^c^	(6.86)	12.01 ^d^	(6.86)	34.72 ^a^	(12.91)	29.19 ^b^	(7.52)	7.00	(7.99)	5.86	(1.13)
1605	*p*-menthadienol (I)	7.35 ^a^	(8.07)	6.36 ^b^	(5.14)	13.13 ^b^	(12.91)	5.19 ^c^	(4.98)	5.62 ^a^	(11.55)	5.71 ^a^	(7.75)	7.76 ^c^	(11.55)	4.45 ^d^	(7.63)	8.26 ^a^	(6.09)	9.76 ^b^	(8.82)	11.41	(4.02)	2.37	(10.29)
1611	*p*-menth-2-en-1-ol	0.33 ^a^	(14.98)	0.35 ^a^	(8.15)	0.21 ^b^	(13.64)	0.69 ^c^	(11.78)	0.47 ^a^	(7.88)	0.37 ^b^	(8.44)	0.36 ^b^	(4.98)	0.37 ^b^	(4.98)	0.25 ^a^	(7.82)	0.36 ^b^	(7.63)	0.27 ^a^	(8.57)	0.09	(8.32)
1646	*p*-menthadienol (II)	8.94 ^a^	(5.47)	7.12 ^b^	(7.63)	7.70 ^b^	(1.40)	3.54 ^c^	(12.97)	4.90 ^a^	(7.63)	4.97 ^a^	(1.16)	4.12 ^b^	(2.63)	3.43 ^c^	(8.44)	6.08 ^a^	(7.88)	5.90 ^a^	(9.68)	5.62 ^ab^	(7.63)	5.19 ^b^	(11.91)
1667	β-citronellol	9.61 ^a^	(2.52)	8.66 ^b^	(3.84)	12.26 ^b^	(7.44)	7.64 ^c^	(2.52)	8.37 ^a^	(0.75)	8.26 ^a^	(7.82)	7.85 ^b^	(0.75)	6.41 ^c^	(9.38)	2.31 ^a^	(7.56)	2.77 ^a^	(6.82)	1.90 ^b^	(9.85)	5.13 ^c^	(13.38)
1786	nerol	14.48 ^a^	(5.81)	15.60 ^a^	(8.63)	10.26 ^b^	(4.90)	12.58 ^c^	(7.63)	7.46 ^a^	(5.60)	7.53 ^a^	(7.04)	7.99 ^b^	(8.60)	2.20 ^c^	(13.45)	5.78 ^a^	(13.15)	5.72 ^a^	(4.07)	1.38 ^b^	(5.05)	0.40 ^c^	(7.31)
1838	geraniol	14.46 ^a^	(13.24)	17.84 ^b^	(7.56)	7.13 ^b^	(8.31)	0.03 ^c^	(0.00)	8.97 ^a^	(10.03)	8.99 ^a^	(7.52)	5.88 ^b^	(10.03)	3.39 ^c^	(15.98)	8.60 ^a^	(9.51)	8.25 ^a^	(13.26)	1.46 ^b^	(3.78)	0.65 ^c^	(5.70)
1834	(*Z*)-carveol	2.28 ^a^	(10.54)	2.51 ^a^	(4.13)	2.42 ^a^	(12.63)	0.03 ^c^	(0.00)	2.77 ^a^	(3.02)	4.77 ^b^	(1.94)	1.53 ^c^	(2.75)	1.64 ^c^	(7.56)	1.10 ^a^	(3.33)	2.05 ^b^	(3.85)	1.70 ^c^	(1.05)	0.91 ^d^	(1.91)
1927	limonyl alcohol	0.73 ^a^	(8.15)	0.75 ^a^	(8.07)	0.59 ^b^	(8.07)	0.61 ^b^	(7.63)	0.58 ^a^	(0.00)	0.82 ^b^	(2.25)	0.62 ^a^	(3.45)	0.37 ^c^	(6.99)	0.21 ^a^	(5.15)	0.38 ^b^	(4.67)	0.20 ^a^	(3.05)	0.12 ^c^	(6.86)
1963	*p*-mentha-dien-9-ol	1.32 ^a^	(9.51)	1.73 ^ab^	(9.31)	1.98 ^b^	(6.99)	1.25 ^b^	(8.63)	3.58 ^a^	(3.43)	3.71 ^a^	(6.50)	2.64 ^b^	(5.43)	2.36 ^b^	(5.82)	0.63 ^a^	(5.05)	1.09 ^b^	(5.14)	0.61 ^a^	(13.73)	0.40 ^c^	(7.52)
1996	perillyl alcohol	2.04 ^a^	(7.18)	2.14 ^a^	(0.03)	2.14 ^a^	(14.74)	0.81 ^c^	(6.86)	1.34 ^a^	(2.42)	2.34 ^b^	(7.52)	0.79 ^c^	(1.38)	0.45 ^d^	(9.51)	0.49 ^a^	(9.59)	2.76 ^b^	(8.44)	0.70 ^c^	(2.37)	0.30 ^d^	(2.87)
2140	eugenol	0.39 ^a^	(8.15)	0.52 ^b^	(3.61)	0.38 ^a^	(10.05)	0.16 ^c^	(8.15)	1.03 ^a^	(3.85)	1.75 ^b^	(7.52)	0.98 ^a^	(3.85)	0.07 ^c^	(5.70)	0.27 ^a^	(7.31)	0.34 ^b^	(7.12)	0.25 ^a^	(7.63)	0.10 ^c^	(12.97)
	Total	160.2		168.41		147.09		88.85		95.42		104.02		93.45		55.62		132.82		176.4		113.48		46.04	
**Monoterpenoid aldehydes**																									
1468	citronellal	3.58 ^a^	(6.33)	3.99 ^a^	(10.40)	2.57 ^b^	(7.76)	0.00 ^c^		1.14 ^a^	(5.82)	1.87 ^b^	(4.07)	1.70 ^c^	(0.71)	0.00 ^d^		2.81 ^a^	(0.60)	3.19 ^b^	(3.35)	0.99 ^c^	(8.82)	0.00 ^d^	
1664	neral	3.66 ^a^	(4.95)	2.88 ^b^	(2.97)	4.74 ^c^	(1.26)	7.15 ^d^	(6.73)	1.33 ^a^	(3.30)	2.84 ^b^	(6.90)	3.37 ^c^	(7.45)	4.11 ^d^	(10.14)	2.20 ^a^	(2.16)	4.29 ^b^	(2.67)	5.95 ^c^	(9.51)	9.94 ^d^	(10.29)
1712	geranial	5.61 ^a^	(1.55)	4.66 ^b^	(6.73)	3.74 ^c^	(6.87)	2.26 ^d^	(5.82)	3.19 ^a^	(5.99)	2.09 ^b^	(7.63)	1.37 ^c^	(9.33)	1.56 ^c^	(7.63)	3.60 ^a^	(14.24)	4.19 ^b^	(11.41)	7.83 ^c^	(19.37)	5.69 ^d^	(0.02)
	Total	12.85		11.53		11.05		9.41		5.66		6.8		6.44		5.87		8.61		11.67		14.77		15.63	
**Chain aldehydes**																									
1284	octanal	12.04 ^a^	(6.73)	9.98 ^b^	(1.51)	5.08	(7.16)	3.39	(5.97)	6.98 ^a^	(8.38)	7.17 ^a^	(17.05)	5.84 ^c^	(7.31)	2.77 ^d^	(9.94)	9.57 ^a^	(14.76)	8.70 ^b^	(1.00)	7.04 ^c^	(2.00)	2.33 ^d^	
1386	nonanal	2.58 ^a^	(8.13)	1.63 ^b^	(6.09)	1.52 ^b^	(13.26)	0.73 ^c^	(7.56)	2.00 ^a^	(1.01)	1.65 ^b^	(3.19)	1.88 ^ab^	(4.86)	1.15 ^c^	(2.42)	2.01 ^a^	(7.63)	2.10 ^a^	(3.61)	2.54 ^b^	(2.16)	0.05 ^c^	
1490	decanal	13.91 ^a^	(7.31)	12.22 ^b^	(4.14)	11.00 ^c^	(6.09)	7.00 ^d^	(8.32)	11.52 ^a^	(3.31)	11.00 ^a^	(5.08)	5.86 ^b^	(1.02)	3.86 ^c^	(1.55)	4.76 ^a^	(7.63)	3.44 ^b^	(7.56)	3.21 ^b^	(5.98)	0.34 ^c^	
	Total	27.83		23.93		18.1		11.12		20.5		20.42		13.58		7.78		16.34		14.24		12.49		2.72	
**Terpene hydrocarbons**																									
1018	α- pinene	13.49 ^a^	(2.77)	13.58 ^a^	(1.48)	21.97 ^b^	(4.98)	8.48 ^c^	(8.03)	11.12 ^a^	(7.75)	19.76 ^b^	(3.31)	21.85 ^c^	(0.98)	9.50 ^d^	(0.98)	17.57 ^a^	(4.60)	17.85 ^a^	(3.05)	11.58 ^c^	(3.00)	10.21 ^d^	(10.81)
1104	β- pinene	5.63 ^a^	(4.52)	6.36 ^b^	(5.79)	7.51 ^c^	(2.76)	5.20 ^d^	(11.55)	4.60 ^a^	(8.13)	5.80 ^b^	(7.91)	7.43 ^c^	(7.18)	2.00 ^d^	(8.13)	11.34 ^a^	(7.56)	9.64 ^b^	(7.82)	13.34 ^c^	(11.22)	1.61 ^d^	(7.76)
1119	sabinene	7.65 ^a^	(13.34)	9.47 ^b^	(2.17)	13.94 ^c^	(8.32)	4.84 ^d^	(4.98)	5.90 ^a^	(10.07)	7.14 ^b^	(2.87)	10.53 ^c^	(7.99)	3.10 ^d^	(2.87)	9.56 ^a^	(3.97)	11.84 ^b^	(4.13)	14.25 ^c^	(6.73)	3.69 ^d^	(8.63)
1209	limonene	1687.15 ^a^	(4.98)	1637.25 ^a^	(1.59)	2055.82 ^b^	(7.31)	986.05 ^c^	(11.78)	1677.04	^a^ (0.79)	1700.36 ^b^	(0.65)	2005.28 ^c^	(0.44)	852.33 ^d^	(7.83)	2107.74 ^a^	(3.65)	2190.26 ^b^	(0.72)	2424.57 ^c^	(8.32)	925.92 ^d^	(7.31)
1237	(*Z*)-b-ocimene	0.31 ^a^	(11.78)	0.48 ^b^	(7.16)	0.75 ^c^	(1.54)	0.15 ^d^	(3.61)	0.23 ^a^	(7.99)	0.32 ^b^	(0.77)	0.43 ^c^	(12.91)	0.14 ^d^	(0.77)	0.45 ^a^	(10.40)	0.41 ^a^	(6.90)	0.59 ^b^	(7.63)	0.00 ^c^	
1254	γ-terpinene	3.19 ^a^	(5.70)	3.46 ^b^	(5.95)	3.77 ^c^	(3.26)	2.21 ^d^	(1.40)	2.08 ^a^	(9.45)	2.34 ^a^	(8.47)	3.37 ^c^	(10.29)	2.21 ^d^	(7.45)	3.14 ^a^	(0.71)	3.99 ^b^	(1.98)	2.72 ^c^	(7.52)	0.78 ^d^	(10.92)
1256	(*E*) b-ocimene	1.73 ^a^	(3.33)	2.28 ^b^	(14.80)	3.26 ^c^	(1.40)	1.59 ^d^	(9.82)	1.39 ^a^	(8.71)	1.69 ^a^	(9.82)	3.23 ^c^	(7.12)	0.76 ^d^	(2.78)	0.60 ^a^	(1.94)	0.49 ^b^	(6.90)	0.64 ^a^	(0.26)	0.41 ^b^	(5.69)
1269	*p*-cymene	0.73 ^a^	(4.11)	0.86^b^	(4.79)	0.91 ^b^	(3.51)	0.69 ^c^	(12.97)	0.42 ^a^	(9.04)	0.46 ^a^	(11.17)	0.46 ^a^	(9.61)	0.20 ^d^	(2.77)	0.38 ^a^	(3.31)	0.40 ^a^	(3.61)	1.09 ^b^	(7.25)	0.25 ^c^	(6.86)
1289	α-terpinolene	2.55 ^a^	(8.38)	2.47 ^a^	(10.14)	3.01 ^b^	(5.14)	1.21 ^c^	(12.32)	2.50 ^a^	(6.93)	2.36 ^a^	(8.22)	3.28 ^b^	(8.32)	1.42 ^c^	(8.32)	3.83 ^a^	(7.99)	4.42 ^b^	(7.44)	3.15 ^c^	(9.07)	0.59 ^d^	(8.62)
1368	alloocimene	0.11 ^a^	(8.13)	0.11 ^a^	(1.75)	0.20 ^b^	(17.91)	0.10 ^c^	(0.98)	0.17 ^a^	(3.55)	0.15 ^a^	(7.41)	0.18 ^a^	(14.93)	0.07 ^d^	(2.70)	0.09 ^a^	(8.47)	0.12 ^a^	(7.82)	0.23 ^c^	(3.13)	0.08 ^a^	(11.55)
1427	*p*-cymenene	0.31 ^a^	(7.99)	0.66	(6.86)	1.68 ^b^	(2.87)	0.75 ^c^	(7.41)	0.23 ^a^	(7.44)	0.10 ^b^	(7.44)	0.62 ^c^	(9.05)	0.00 ^d^		0.35 ^a^	(7.88)	1.35 ^b^	(6.40)	1.48 ^b^	(5.70)	0.19 ^c^	(3.61)
1521	β-cubenene	0.35 ^a^	(7.52)	0.69 ^b^	(6.73)	0.90 ^c^	(7.16)	0.42 ^d^	(7.31)	1.23 ^a^	(3.33)	0.54 ^b^	(3.33)	0.83 ^c^	(7.63)	1.28 ^a^	(7.18)	0.44 ^a^	(13.26)	0.41 ^a^	(6.09)	1.20 ^c^	(6.93)	0.28 ^d^	(8.15)
1578	*trans*Caryophyllene	2.14 ^a^	(6.86)	2.96 ^b^	(5.82)	6.62 ^b^	(2.59)	1.77 ^c^	(6.86)	4.09 ^a^	(11.62)	4.45 ^ab^	(4.60)	4.77 ^b^	(1.36)	2.01 ^c^	(6.09)	2.75 ^a^	(8.13)	3.59 ^b^	(1.57)	7.50 ^c^	(10.89)	2.14 ^d^	(9.38)
1703	β-selinene	2.52 ^a^	(0.36)	3.30 ^b^	(1.49)	4.43 ^c^	(7.21)	0.30 ^d^	(8.15)	4.58 ^a^	(10.72)	4.13 ^a^	(11.69)	5.81 ^b^	(6.44)	2.39 ^c^	(14.76)	2.30 ^a^	(3.26)	2.90 ^b^	(7.52)	4.67 ^c^	(4.65)	1.62 ^d^	(7.88)
	Total	1727.86		1683.49		2124.68		1013.76		1715.58		1749.1		2068.09		877.72		2190.64		2236.98		2487.17		947.84	
**Terpene Esters**																									
1546	linalyl acetate	0.82 ^a^	(6.86)	0.64 ^b^	(0.22)	0.41 ^c^	(8.32)	0.00 ^d^		0.09 ^a^	(12.91)	0.11 ^a^	(10.00)	0.01 ^b^	(1.00)	0.00 ^c^		1.13 ^a^	(13.65)	0.89 ^b^	(5.06)	0.41 ^c^	(4.98)	0.29 ^d^	(1.01)
1570	bornyl acetate	0.48 ^a^	(8.63)	0.26 ^b^	(6.99)	0.24 ^b^	(5.70)	0.00 ^c^		0.38 ^a^	(1.09)	0.36 ^b^	(6.53)	0.34 ^a^	(1.02)	0.12 ^b^	(14.76)	0.19 ^a^	(1.82)	0.25 ^b^	(1.00)	0.01 ^c^	(0.00)	0.01 ^d^	(0.00)
1644	citronelly acetate	0.23 ^a^	(6.86)	0.20 ^a^	(10.05)	0.12 ^b^	(3.61)	0.11 ^b^	(9.82)	0.16 ^a^	(1.40)	0.14 ^b^	(1.40)	0.00 ^b^		0.00 ^b^		1.41 ^a^	(5.70)	1.33 ^a^	(6.99)	1.41 ^a^	(8.44)	0.68 ^b^	(2.15)
1673	terpinyl acetate	3.80 ^a^	(13.65)	3.49 ^a^	(13.26)	1.08 ^b^	(10.14)	0.82 ^c^	(12.97)	1.36 ^a^	(0.00)	1.30 ^a^	(0.00)	1.16 ^b^	(7.12)	0.64 ^c^	(6.73)	3.22 ^a^	(5.70)	2.51 ^b^	(2.63)	0.45 ^c^	(8.47)	0.57 ^d^	(8.13)
1711	geranyl acetate	0.51 ^a^	(4.30)	0.49 ^a^	(6.86)	0.30 ^b^	(10.14)	0.17 ^c^	(7.63)	1.38 ^a^	(7.42)	0.49 ^b^	(5.70)	0.37 ^c^	(8.71)	0.27 ^d^	(11.92)	0.53 ^a^	(10.07)	0.55 ^a^	(6.86)	0.51 ^a^	(7.75)	0.26 ^b^	(7.82)
1742	neryl acetate	0.72 ^a^	(12.55)	0.77 ^a^	(3.61)	0.39 ^b^	(3.64)	0.23 ^c^	(8.15)	0.68 ^a^	(2.46)	0.66 ^a^	(2.68)	0.50 ^b^	(0.00)	0.32 ^c^	(10.68)	1.04 ^a^	(3.92)	1.09 ^a^	(1.82)	0.80 ^b^	(2.90)	0.34 ^d^	(5.09)
	Total	6.56		5.72		2.54		1.39		4.05		3.08		2.39		1.36		7.72		6.38		3.59		2.15	
**C6 Alcohols**																									
1350	1-hexanol	10.21 ^a^	(12.97)	9.83 ^a^	(0.31)	1.04 ^b^	(8.13)	0.00 ^c^		2.76 ^a^	(1.01)	2.01 ^b^	(1.36)	0.76 ^c^	(36.35)	0.00 ^d^		1.73 ^a^	(6.86)	1.69 ^a^	(9.68)	0.67 ^b^	(12.91)	0.32 ^c^	(8.63)
1381	(*Z*)-3-hexen-1-ol	3.55 ^a^	(0.98)	3.39 ^a^	(2.08)	0.26 ^b^	(10.07)	0.00 ^c^		1.30 ^a^	(8.19)	1.07 ^a^	(5.98)	0.19 ^b^	(0.00)	0.00 ^c^		0.58 ^a^	(8.71)	0.52 ^a^	(8.63)	0.14 ^b^	(4.98)	0.00 ^c^	
	Total	13.76		13.22		1.30		0,00		4.06		3.08		0.95		0.00		2.31		2.11		0.81		0.33	

KI: Kovats index; a, b, c, d: Different letters in the same row denote significant difference according to the Student-Newman-Keuls test (*p* < 0.05).

**Table 2 molecules-26-05928-t002:** Mean values of phenolic acids (mg. g^−1^ DW) and relative standard deviation (RSD) in fresh extracts, oven-dried at 50 °C (OD-50 °C), oven-dried at 70 °C (OD-70 °C), and freeze-dried (FD).

Phenolic Acids	Navelina Peel (*Citrus sinensis* L. Osbeck cv. “Navelina”)								Salustriana Peel (*Citrus sinensis* L. Osbeck cv. “Salustriana”)								Sanguina Peel (*Citrus x sinensis* var. “Sanguina”)							
	Fresh	RSD	OD 50 °C	RSD	OD 70 °C	RSD	FD	RSD	Fresh	RSD	OD 50 °C	RSD	OD 70 °C	RSD	FD	RSD	Fresh	RSD	OD 50 °C	RSD	OD 70 °C	RSD	FD	RSD
Gallic acid	0.03 ^a^	(2.40)	0.03 ^a^	(10.10)	0.14 ^b^	(6.24)	0.29 ^c^	(2.40)	0.03 ^a^	(10.48)	0.04 ^a^	(12.41)	0.13 ^b^	(9.88)	0.29 ^c^	(4.88)	0.03 ^a^	(6.73)	0.03 ^a^	(2.40)	0.15 ^b^	(3.87)	0.40 ^c^	(3.45)
p-Hydroxybenzoic acid	0.02 ^a^	(7.79)	0.03 ^a^	(9.43)	0.07 ^b^	(8.88)	0.10 ^c^	(7.44)	0.02 ^a^	(11.49)	0.03 ^a^	(10.48)	0.06 ^b^	(7.95)	0.09 ^c^	(13.70)	0.05 ^a^	(11.79)	0.06 ^a^	(9.43)	0.14 ^b^	(7.88)	0.20 ^c^	(7.44)
Vanillic acid	0.07 ^a^	(6.24)	0.07	(1.00)	0.26 ^b^	(8.32)	0.73 ^c^	(7.75)	0.06 ^a^	(9.99)	0.06 ^a^	(14.08)	0.22 ^b^	(8.64)	0.64 ^c^	(4.65)	0.14 ^a^	(6.24)	0.15 ^a^	(1.00)	0.54 ^b^	(8.32)	1.54 ^c^	(7.75)
Chlorogenic acid	0.17 ^a^	(8.32)	0.19 ^a^	(7.44)	1.08 ^b^	(7.24)	1.94 ^c^	(5.48)	0.14 ^a^	(12.31)	0.17 ^a^	(10.64)	0.93 ^b^	(7.83)	1.67 ^c^	(6.35)	0.36 ^a^	(8.32)	0.40 ^a^	(7.44)	2.27 ^b^	(7.24)	4.08 ^c^	(5.48)
Caffeic acid	0.30 ^a^	(4.71)	0.38 ^a^	(1.75)	0.75 ^b^	(8.54)	1.02 ^c^	(10.45)	0.26 ^a^	(4.54)	0.31 ^a^	(4.68)	0.65 ^b^	(6.46)	0.90 ^c^	(6.09)	0.63 ^a^	(4.71)	0.85 ^a^	(1.75)	1.57 ^b^	(8.54)	2.14 ^c^	(10.45)
*p*-Coumaric acid	0.31 ^a^	(8.40)	0.38 ^a^	(1.89)	0.43 ^b^	(3.29)	0.89 ^c^	(3.99)	0.27 ^a^	(9.92)	0.29 ^a^	(14.63)	0.38 ^b^	(5.42)	0.75 ^c^	(6.88)	0.66 ^a^	(8.40)	0.67 ^a^	(0.89)	0.91 ^b^	(3.29)	1.87 ^c^	(3.99)
Ferulic acid	0.45 ^a^	(2.37)	0.46 ^a^	(0.77)	1.36 ^b^	(1.57)	1.38 ^c^	(4.63)	0.38 ^a^	(5.89)	0.38 ^a^	(4.73)	1.17 ^b^	(2.27)	1.22 ^c^	(3.39)	0.94 ^a^	(2.37)	0.97 ^a^	(0.77)	2.86 ^b^	(1.57)	2.90 ^c^	(4.63)
Sinapic acid	0.40 ^a^	(3.54)	0.41 ^a^	(8.73)	1.0 5 ^b^	(8.80)	1.62 ^c^	(8.32)	0.35 ^a^	(5.00) ^a^	0.35 ^a^	(10.64)	0.91 ^b^	(9.92)	1.36 ^c^	(5.35)	0.84 ^a^	(3.54)	0.85 ^a^	(8.73)	2.20 ^b^	(8.80)	3.41 ^c^	(8.32)
Total acids	1.75 ^a^		1.94 ^a^		5.11 ^b^		7.94 ^c^		1.50 ^a^		1.63 ^a^		4.44 ^b^		6.92 ^c^		3.66 ^a^		3.98 ^a^		10.63 ^b^		16.54 ^c^	

a, b, c: Different letters in the same row denote a significant difference according to the Student-Newman-Keuls test, at *p* < 0.05. DW: dry weight basis.

**Table 3 molecules-26-05928-t003:** Spectral data of flavonoids in orange peel extracts.

Tentative Identification	Retention Time (min)	UVmax (nm)	MS [M − H]^−^ (*m*/*z*)	MS [M + H]^+^ (*m*/*z*)	Products Ions (*m*/*z*)
Rutin [89]	11.43	256, 286, 351	611		300.8, 342.8
Narirutin [87,90]	13.58	217, 284, 331	579		271, 151
Naringin [87,90]	18.02	224, 283,331	579		459, 271
Hesperidin [87,90]	20.19	225, 284, 328	609		301, 198
Naringenin [87,89,90]	24.65	226, 284, 325		273	153
Hesperetin [89]	26.57	225, 285, 329		303	285
Sinensetin [89,90]	28.04	243, 264, 333		373	358, 343, 312
Quercetogenin [89]	31.98	250, 272, 335		403	388, 373
Nobiletin [89,90]	35.46	248, 268, 334		403	388, 373
Tangeretin [87,89,90]	36.54	271, 322		373	358, 343, 325, 297

**Table 4 molecules-26-05928-t004:** Mean concentration of flavonoids (mg. g^−1^ DW) in Navelina, Salustriana, and Sanguina peels, both fresh and after drying treatment: oven-dried (50 °C), oven-dried (70 °C), and freeze-dried (FD).

	Navelina Peel (*Citrus sinensis* L. Osbeck cv. “Navelina”)								Salustriana Peel (*Citrus sinensis* L. Osbeck cv. “Salustriana”)								Sanguina Peel (*Citrus x sinensis* var. “Sanguina”)							
	Fresh	RSD	Oven 50 °C	RSD	Oven 70 °C	RSD	FD	RSD	Fresh	RSD	Oven 50 °C	RSD	Oven 70 °C	RSD	FD	RSD	Fresh	RSD	Oven 50 °C	RSD	Oven 70 °C	RSD	FD	RSD
Flavonoids Rutin	3.24 ^a^	(0.24)	2.62 ^b^	(9.87)	1.55 ^c^	(0.45)	3.11 ^a^	(1.81)	3.32 ^a^	(2.21)	2.62 ^b^	(1.62)	1.11 ^c^	(12.49)	2.98 ^a^	(2.98)	5.92 ^a^	(3.21)	3.94 ^b^	(2.51)	2.47 ^c^	(7.18)	5.37 ^a^	(2.98)
Narirutin	28.10 ^a^	(3.11)	21.56 ^b^	(4.36)	18.44 ^c^	(11.92)	27.87 ^a^	(2.51)	26.87 ^a^	(3.25)	19.61 ^b^	(0.35)	17.01 ^c^	(7.75)	23.92 ^d^	(1.57)	47.83 ^a^	(3.25)	39.18 ^b^	(0.66)	31.38 ^c^	(2.53)	42.57 ^d^	(1.57)
Hesperidin	124.73 ^a^	(3.63)	92.97 ^b^	(4.21)	84.22 ^c^	(0.55)	113.70 ^d^	(2.73)	124.00 ^a^	(3.65)	92.06 ^b^	(0.52)	92.23 ^c^	(4.24)	114.97 ^d^	(2.75)	202.32 ^a^	(3.65)	165.21 ^b^	(3.81)	135.49 ^c^	(0.51)	187.59 ^d^	(2.75)
Neohesperidin	11.62 ^a^	(1.30)	8.53 ^b^	(3.84)	7.02 ^c^	(0.26)	10.90 ^d^	(0.03)	10.39 ^a^	(1.45)	7.80 ^b^	(4.87)	6.79 ^c^	(0.27)	9.67 ^a^	(5.03)	18.49 ^a^	(1.45)	15.49 ^b^	(0.80)	13.87 ^c^	(0.24)	17.72 ^a^	(9.03)
Naringin	12.15 ^a^	(2.08)	7.25 ^b^	(0.12)	5.80 ^c^	(1.35)	10.40 ^d^	(2.84)	10.92 ^a^	(2.32)	6.57 ^b^	(1.19)	5.02 ^c^	(0.18)	10.14 ^a^	(3.22)	19.44 ^a^	(2.32)	12.70 ^b^	(1.10)	10.71 ^c^	(0.15)	16.32 ^d^	(3.22)
Hesperetin	5.55 ^a^	(0.29)	6.98 ^b^	(0.93)	8.87 ^c^	(0.72)	8.05 ^d^	(2.76)	4.61 ^a^	(0.15)	5.04 ^b^	(1.19)	7.53 ^c^	(6.39)	6.09 ^d^	(2.91)	8.21 ^a^	(0.15)	10.09 ^b^	(0.78)	12.51 ^c^	(3.22)	11.20 ^d^	(2.27)
Naringenin	3.77 ^a^	(2.63)	4.73 ^b^	(5.54)	7.42 ^c^	(0.10)	6.13 ^d^	(2.08)	3.64 ^a^	(2.32)	4.19 ^b^	(0.62)	6.93 ^c^	(0.71)	5.19 ^d^	(1.70)	3.57 ^a^	(0.15)	4.90 ^b^	(0.87)	6.48 ^c^	(2.32)	5.44 ^d^	(3.22)
Sinensetin	1.64 ^a^	(4.94)	1.61 ^a^	(0.60)	2.28 ^c^	(4.44)	2.05 ^d^	(0.58)	0.41 ^a^	(12.66)	0.38 ^b^	(2.53)	0.95 ^c^	(3.87)	0.82 ^d^	(0.70)	0.73 ^a^	(11.66)	0.69 ^a^	(2.53)	1.69 ^c^	(3.87)	1.46 ^d^	(0.70)
Quercetogenin	2.79 ^a^	(3.47)	3.35 ^b^	(5.04)	4.98 ^c^	(0.14)	4.26 ^d^	(2.03)	2.06 ^a^	(12.56)	3.02 ^b^	(1.87)	4.18 ^c^	(6.12)	3.70 ^d^	(2.43)	3.67 ^a^	(12.56)	3.04 ^a^	(2.43)	7.11 ^c^	(0.17)	6.36 ^d^	(2.42)
Nobiletin	8.53 ^a^	(3.84)	9.02 ^b^	(0.21)	11.91 ^c^	(2.17)	10.90 ^d^	(0.03)	7.95 ^a^	(11.13)	9.28 ^b^	(0.84)	12.83 ^c^	(5.82)	11.17 ^d^	(2.64)	16.51 ^a^	(0.84)	15.46 ^a^	(1.90)	26.40 ^c^	(5.82)	19.88 ^d^	(2.64)
Tangeretin	3.37 ^a^	(1.54)	4.23 ^b^	(6.63)	6.34 ^c^	(2.48)	5.53 ^d^	(2.15)	3.60 ^a^	(0.32)	4.00 ^b^	(1.88)	6.12 ^c^	(4.85)	5.04 ^d^	(5.20)	6.41 ^a^	(0.32)	7.12 ^b^	(1.88)	10.89 ^c^	(4.85)	8.96 ^d^	(5.20)
TPC (mg GAE·g^−1^ DW)	22.75 ^a^	(10.88)	30.50 ^b^	(6.96)	42.00 ^c^	(10.10)	53.29 ^d^	(1.09)	19.55 ^a^	(3.26)	27.16 ^b^	(1.87)	31.27 ^c^	(10.27)	45.09 ^d^	(3.42)	41.66 ^a^	(1.17)	64.36 ^b^	(6.96)	83.15 ^c^	(1.45)	117.15 ^d^	(4.64)
TFC mg (QE·g^−1^ DW)	12.64 ^a^	(1.23)	16.94 ^b^	(1.04)	23.33 ^c^	(4.13)	29.61 ^d^	(6.96)	10.86 ^a^	(9.88)	15.09 ^b^	(3.12)	19.37 ^c^	(4.43)	25.05 ^d^	(7.25)	23.14 ^a^	(5.77)	30.7 5 ^b^	(3.45)	35.19 ^c^	(1.86)	46.00 ^d^	(0.44)

a, b, c, d: Different letters in the same row denote a significant difference according to the Student-Newman-Keuls test, at *p* < 0.05. TPC total phenolic compounds. GAE—gallic acid equivalent; TFC—total flavonoids content; QE—quercetin equivalent.

**Table 5 molecules-26-05928-t005:** Antioxidant activity (mg trolox·g^−1^ DW) and relative standard deviations (RSD) for fresh and dried Navelina, Sanguina, and Salustriana orange peels.

	Navelina Peel (*Citrus sinensis* L. Osbeck cv. “Navelina”)								Salustriana Peel (*Citrus sinensis* L. Osbeck cv. “Salustriana”)								Sanguina Peel (*Citrus x sinensis* var. “Sanguina”)							
	Fresh	RSD	Oven 50 °C	RSD	Oven 70 °C	RSD	FD	RSD	Fresh	RSD	Oven 50 °C	RSD	Oven 70 °C	RSD	FD	RSD	Fresh	RSD	Oven 50 °C	RSD	Oven 70 °C	RSD	FD	RSD
DPPH	44.82 ^a^	(11.94)	56.30 ^b^	(2.54)	64.96 ^c^	(4.43)	71.57 ^d^	(0.84)	33.62 ^a^	(11.94)	42.22 ^b^	(2.54)	47.72 ^c^	(4.52)	53.68 ^d^	(0.84)	67.24 ^a^	(11.94)	80.45 ^b^	(0.95)	93.38 ^c^	(1.53)	107.36 ^d^	(0.84)
FRAP	26.89 ^a^	(11.94)	33.78 ^b^	(2.54)	42.92 ^c^	(3.58)	49.49 ^d^	(6.16)	18.83 ^a^	(11.94)	23.65 ^b^	(2.54)	30.04 ^c^	(3.58)	34.64 ^d^	(6.16)	38.73 ^a^	(11.94)	48.64 ^b^	(2.54)	63.36 ^c^	(0.01)	71.26 ^d^	(6.16)
ABTS	144.94 ^a^	(2.44)	213.62 ^b^	(2.65)	253.35 ^c^	(1.95)	313.07 ^d^	(0.43)	122.48 ^a^	(0.56)	159.08 ^b^	(1.87)	196.09 ^c^	(0.01)	239.85 ^d^	(3.23)	202.69 ^a^	(0.61)	283.48 ^b^	(0.97)	364.58 ^c^	(5.54)	569.23 ^d^	(0.43)

a, b, c, d: Different letters in the same row denote a significant difference according to the Student-Newman-Keuls test, at *p* < 0.05. DW—dry weight basis. FD—freeze-dried.

## Data Availability

The data presented in this study are available on request from the corresponding author.

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
