# Peer review of "Pressurized Extraction as an Opportunity to Recover Antioxidants from Orange Peels: Heat treatment and Nanoemulsion Design for Modulating Oxidative Stress"

_molecules, 2021, doi:10.3390/molecules26195928_

Round 1

Reviewer 1 Report

To research on food wastes as a source of bioactive phytochemicals is a growing topic. This work is interesting and may contribute to it. However, it shows many inconsistencies between the methodology and results section; thus, the discussion and conclusions could be not adequate. The authors try to determine if the heat treatment for drying the orange peel from three varieties influences the profile of terpene and phenolic compounds since they are secondary metabolites recognized as bioactive compounds; however, results don´t respond clearly and concisely to this question.  Moreover, there are several data described without showing the application of statistical analysis, so it can be confusing to try to understand what the authors mean.  In addition, results should be described in a logical sequence. Discussion and conclusions should be rewritten based on the statistical analysis of the results. All the abbreviations must be described by the first time. Several comments are described for authors for improving the quality of their work.

TITLE

Can be improved more concisely, based on the assumption that heat treatment affect the profile/amount of terpene and phenolic compounds, secondary metabolites able to modulate oxidate stress in a cellular line when they are incorporated in a nanoemulsion

ABSTRACT

Line 15 delete point between potential and from.

In line 18 is described: remain practically unaffected among fresh and freeze-dried orange peels. However, lines 14-15 is describe oven-drying and freeze-drying, but it is not described fresh extraction, maybe you should be described that fresh extraction was the control.

Was the profile of terpene derivatives and phenolic compounds and antioxidant potential differences among the source of peel wastes?. Maybe it should be described in the abstract. Maybe the abstract structure must be improved more concisely, focusing on differences between samples, and between treatments irrespective of the type of peel waste. The best results can be briefly described. The abstract must show if all the samples were tested for nanoemulsion formulations, or if it was selected the sample showing the best results based on antioxidant potential.

In line 26 is describe pressurized extracts, which weren´t described before. Even, "pressurized extraction" is described from the title; it must show consistency. 

Maybe about nanoemulsions information (lines 23-26), you should describe the importance of testing surfactants, co-surfactants for delivery of a nanoemulsion capable to protect antioxidants compounds, since the title of the work describes "design".

INTRODUCTION

Line 59. Correct CO2 for CO2

Line 87. Use [54,55,56]

Line 96-97  Describe briefly information about the varieties of orange, which allow justifying their use in the work.

Material and Method

This section must be substantially improved, since there are many omitted data, such as how was determined the mean and RSD showed in Tables 1, 2, and 4. How was determine the percentage in Figure 1, etc?

 Line 121 describe how was carried out the freeze-dried

Section 2.3. Describe which was samples tested for this experiment; biomass from fresh, dried by the oven, and freeze-dried.

Line 142. Describe the name of the pure standards

In line 123 and line 149 is described the procedure used in ASE for secondary metabolite extraction, so, why is line 123 described as "isolation"?…

Section 2.3 and 2.4 are similar, but 2.4 is described as 2.4.1 for extraction and 2.4.2. for HPLC-MS analysis. Homogenize both sections.

Line 184-185. Describe the name of the pure standards

Line 192 Correct: mg naringin equivalents g-1

Line 209: V-Vis spectra are correct?

Line 213: it is described as capillary exit offset. However, in line 178-182 was not described

Line 221-222. It is usual to express the total phenolic content as mg GAE per gram of dry biomass or dried extract; it was done?. Line 230-231 was described for total flavonoids content.

Line 2.4.1. It is important to describe which procedure was made for biomass: fresh? Is the oven-dry? Or freezing dry?. Same comment for the procedure described in lines 123-125

Line 119. For fresh biomass is describe that the peel portion is directly analyzed. It might result confusing thus extraction in ASE was omitted?, Analyzed seems peels were directly analyzed for HPLC/GC-MS. Maybe in this section must be described that all biomass resulting from i), ii) and iii) for every variety of orange was extracted by ASE and the resulting extracts were analyzed by HPLC-MS, GC-MS, and phenolics and antioxidant potential quantification.

Line 242, 258. Describe if the results are expressed regarding biomass or extract dry weight

Line 276-277. Describe if the selection of extract showing the highest antioxidant potential depended on the orange variety, as well as the procedure tested for drying the peels (fresh, oven-dry, freezing dry)

RESULTS AND DISCUSSION

In general terms, this section doesn´t show a correct consistency with the information described in the methodology, and, it must be shown in a sequential way regarding the procedure described in the methodology. The paper in this section must show clearly how the heat-treatment could modify the profiles of terpene and phenolic compounds among the three varieties. Following, results should show clearly how the heat-treatment could modify the concentration of phenolic contents as well the antioxidant potential among the three varieties since the highest values observed were used as a criterium to select the extract used to deliver a nanoemulsion. Finally, show how the nanoemulsion confers protection to cells. Some comments are:

Table 1 can be improved in its structure. In section 2.3 it describes the use of standards when they were available. So, a calibration curve was built for every standard to obtain a quantitative analysis?... however, it is not clear how was made the semiquantitative analysis?, or if the standard was unavailable, how to determine the amount of the compound; since in table 1 all the data are expressed on micrograms/g…?... multivariate analysis is more recommended when no standards are available through using the abundance of every peak and to use Fold Change, and Variable importance Projection allow show in a more concise way which are the main differences caused by the treatment of the biomass peel.  What does it mean “KI”? Use only a title on top of the table.

In line 336, preferably, describe FD as freeze-dried

Line 340. The location of “Figure 1” Is incorrect, because this figure “doesn’t explain based on the low processing temperatures….”. It must describe before the point, in line 338

Figure 1 shows the percentage regarding all the secondary metabolites detected?. It should be described in the methodology section. Once all compounds were putatively identified, they were separated as families and the percentages were assumed?.... However, the assumption that the volatile profile found between Fresh and Freeze-dried peels is similar was not related only to the "presence" of the compounds, but also an abundance of peaks or concentration should be analyzed. This could be better explained with fold change multivariate analysis. In the caption of the figure, avoid the use of “behavour"/behavior; the figure shows percentages of family compounds detected regarding the total of detected compounds, isn´t it?

Line 344-346: Were is observed the described results?. In table 1 did you carry on a sum of the total VOCs concentration? it is not shown how was it determined? The methodology doesn´t describe this information. Line 345 and 346, Navelina is twice mentioned. Line 346-347, where can be observed that limonene is the major aroma constituent; according to the reports mention in several works, can establish a comparison regarding the concentration observed in this work regarding other works?

It is confusing the idea shown in lines 349-351 because in the previous paragraph was mentioned limonene as a major aroma constituent on percentage, and after, is described the terpene hydrocarbons comparison regarding concentration

Figure 6  Show data derived from statistical analysis, including the test used and its significance. However, it is not clearly described in some other tables or figures, so, when the statistical analysis was not realized, how can describe the results when a comparison was done for example from table 1?.

Supplementary material about the mass spectrum obtained from the HPLC-MS, GC-MS analysis must be considered.

CONCLUSIONS

Improve, it must be concise according to the results, hypothesis, and aim of this work

Reviewer 2 Report

The manuscript “Pressurized Extraction as An Opportunity to Recovery Antioxidants from Orange Peels. Design of Nanoemulsions for Modulating Oxidative Stress” deals with the extraction of bioactive compounds from Navelina, Salustriana and Sanguina peels wastes, for food applications. This is a very accurate study, clearly presented and well written. The publication is recommended. However, some minor revisions are required, as follows:

- Some typing errors are present. Please, check and correct them.

- Introduction. The state of the art related to the extraction methods can be enlarged highlighting advantages and disadvantages of the various techniques. As an example, see this recent review: Baldino et al., Supercritical fluid technologies applied to the extraction of compounds of industrial interest from Cannabis sativa L. and to their pharmaceutical formulations: A review, Journal of Supercritical Fluids, 2020, 165, 104960.

- Use “mL” instead of “ml”.

- Use “MPa” instead of “psi”.

- Results. A comparison on the quantity and purity of the bioactive extracts found in this work with the results present in the literature, on the same vegetable matrices but using different extraction methods should be interesting to add in this section.

- The quality of the figures has to be improved.

Author Response

Please see the attachmet

Round 2

Reviewer 1 Report

The new version has been substantially improved; thus, it can be accepted. However, some corrections must be done according to the following comments.

Table 1. Different letters in the same column denote significant differences. However, it means that the comparison was made between different compounds? it seems wrong because the objective of this statistical analysis is to compare the effect of the heat treatment on the amount of every particular compound. Maybe the letter is considered regarding the row instead of the columns for every variety. In addition, it is not logical that in some data for example in the column Mean Oven 50ºC for Navelia peel, the mean for α-pinene of 27.97b shows the same letter as limonene 2055.82 b. Review

Line 459 Review the units for concentration: 1000 mg/g… it seems wrong

Figure 2. Title for Y-axis must be identified as total phenolic content (mg GAE g-1). Erase the title of the graphic

Take care in all the manuscript for using the super index to denotes units at -1, such as mg GAE g-1 described in 531-534 lines

Lines 542-543. What does it mean  "FD-Na-vel peels by FD in comparison…."?, is Navel another orange variety?; in this idea is compared FD vs oven-dried, thus maybe "FD" from "FD-Na-vel " can be erased

Line 565 Use uppercase for “Sanguina”

Line 578. Write adequately the scientific name of Citrus sinensis; avoid using uppercase for the species

In some cases, the results from secondary metabolite contents are described using a diagonal, while in other cases it is used g-1. Homogenize. As well, in some cases, the results are described per gram (/g or g-1) following DW or omitting DW. Homogenize.

Lines 583-586. Review the data, since 815 mg/g is a very high concentration, which could be similar to 0.815 gram of caffeic acid regarding a gram (in all the manuscript should be clear if this gram is regarding the extract), which means an extract comprehends in approximately 80% of a sole compound. Regarding this work, line 585 describes "an intermediate concentration of 630 mg/g DW in Sanguina peels", which is confusing since it seems that this value also refers to caffeic acid, which is not described in Table 2. In guest that the units must be described as micrograms/g. since in table 1 the data are shown in micrograms/g, table 2 and Figure 3 must also show the results in micrograms/g

Table 2. Similar to in table 1, it seems that the letter are described regarding the same row.

Line 602. Review if it is correct to describe the results regarding mg/g

Figure4 must show the results from the statistical analysis

Line 632. Table 3 doesn´t show the quantification data, this table shows parameters related to HLPC-MS and LC-MS

Line 637. Review if the units mg/g is correct

Line 640. Use cursives for the scientific name

Line 665. What does it mean: m/g?

Line 593 indicates the units of concentration: micrograms/g?

Table 4. It seems that letters are shown regarding a row. However. For rutin, it is not logical that FD had simultaneously the letters a,d (3.03 a,d). Review carefully how the letters are assigned according to the pos-hoc test.

Same comment in section 3.6. Are you sure that the results must be shown on mg/g?

Lines 733 and 744. Homogenize if it is used mg Trolox g-1 or mg Trolox/g DW. Review it in all the manuscript

Line 766. Pearson correlation is equal to R

Table 5. The same, it seems the letters are shown regarding the same row

Figure 5. It is important to show the statistical value obtained for p for every test run for Pearson analysis

Figure 6. The graphic is used 5. a, but in the caption, it was not always described. Homogenize. Review the order of the letter showing statistical differences in 5.a, 5. b and 5.c… It seems that "b" must be "c" and "c" must be "b". In 5.a, 5.b and 5.c  "a" correspond to the highest value, but in 5.d "a" corresponds to the lowest value. Homogenize

Line 899-902. Please, indicates statistical differences using letters in Figure 8.

Integrate the conclusion shown between 977-979, and 983-985, since they are bases on freeze-dried treatment

Author Response

thank you in advance
